# Identification of blowing snow particles in images from a multi-angle snowflake camera

Mathieu Schaer[1], Christophe Praz[1], and Alexis Berne[1]

[1] Environmental Remote Sensing Laboratory, École Polytechnique Fé dérale de Lausanne, Switzerland

**Correspondence:** Alexis Berne (alexis.berne@epfl.ch)

**Abstract.** A new method to automatically discriminate between hydrometeors and blowing snow particles on Multi-Angle Snowflake Camera (MASC) images is introduced. The method uses four selected descriptors related to the image frequency, the number of particles detected per image as well as their size and geometry to classify each individual image. The classification task is achieved with a two components Gaussian Mixture Model fitted on a subset of representative images of each class from field campaigns in Antarctica and Davos, Switzerland. The performance is evaluated by labeling the subset of images on which the model was fitted. An overall accuracy and Cohen's Kappa score of 99.4 and 98.8%, respectively, is achieved. In a second step, the probabilistic information is used to flag images composed of a mix of blowing snow particles and hydrometeors, which turns out to occur frequently. The percentage of images belonging to each class from an entire austral summer in Antarctica and during a winter in Davos, respectively, are presented. The capability to distinguish precipitation, blowing snow and a mix of those in MASC images is highly relevant to disentangle the complex interactions between wind, snowflakes and snowpack close to the surface.

## 1 Introduction

Over snow covered regions, ice particles can be lifted from the surface by the wind and suspended in the atmosphere. Wind-driven snow transport is ubiquitous in the cryosphere: over complex terrain (e.g. Winstral et al., 2002; Mott and Lehning, 2010), over tundra/prairies (e.g. Pomeroy and Li, 2000) and over polar ice sheets (e.g. Bintanja, 2001; Déry and Yau, 2002; Palm et al., 2011). Wind-driven snow transport must be taken into account to obtain accurate estimates of the mass balance and radiative forcings at the surface (e.g. Gallée et al., 2001; Lesins et al., 2009; Scarchilli et al., 2010; Yang et al., 2014). In mountainous regions, wind-transported snow also creates local accumulations and irregular deposits, being a critical factor influencing avalanche formation (e.g. Schweizer et al., 2003). Quantifying snow transport during snowfall events and subsequent periods of strong winds is essential for local avalanche prediction (e.g. Lehning and Fierz, 2008). In the context of climate change, the mass balance of the Antarctic ice sheet is of increasing relevance due to its impact on sea level rise (Shepherd et al., 2012). The sustained katabatic winds in Antarctica generate frequent blowing snow events, that remove a significant amount of new snow

through transport and sublimation. Wind-transported snow is hence an important factor to take into account when considering Antarctic mass balance (e.g. Déry and Yau, 2002; Scarchilli et al., 2010; Lenaerts and van den Broeke, 2012; Das et al., 2013). Blowing snow is also an important process for the mass balance of the Greenland ice sheet (e.g. Box et al., 2006).

Ice particles moving at the snow surface belong to one of the three main types of associated motion: creep, saltatation and suspension (e.g. Kind, 1990). Given the fact that the observations used in the present study were collected about 3 m above the ground (or snow surface) level, the term "blowing snow" hereinafter refers to wind-suspended ice particles.

Blowing snow is challenging to measure and characterize. Various approaches have been proposed to monitor blowing snow at ground level: mechanical traps, nets, photoelectric or acoustic sensors, photographic systems (Leonard et al., 2012; Kinar and Pomeroy, 2015). Although not specifically designed for blowing snow, present weather sensors have been shown to be valuable to monitor drifting and blowing snow fluxes (e.g. Bellot et al., 2011). Remote sensing, and lidar systems in particular, have recently been used to characterize the occurrence and depth of blowing snow layers, either from space (Palm et al., 2011) or near ground-level (Gossart et al., 2017). Suspended ice particles are under the influence of the gravitational force, proportional to the size cubed while the drag force is proportional to the area (size squared). With a greater area to mass ratio, smaller particles are thus more likely to be lifted in the suspension layer. A comparison of ten different studies of measured and simulated particle size distributions of blowing snow, reveals mean diameters at heights above $0.2\,\mathrm{m}$ ranging from $50$ to $160\,\mathrm{\mu m}$ (Gordon and Taylor, 2009).

Blowing snow may also contaminate precipitation observations collected by ground-based sensors, frequently in Antarctica (e.g. Nishimura and Nemoto, 2005; Gossart et al., 2017) where winds are strong and frequent, but also in snowy regions in general (Rasmussen et al., 2012; Naaim-Bouvet et al., 2014; Scaff et al., 2015). The issue of snowfall measurement is complex and WMO promoted intercomparison projects to evaluate various sensors and define standards set-ups and protocols over the last two decades, as illustrated in (Goodison et al., 1998) and the recent SPICE project (http://www.wmo.int/pages/prog/www/IMOP/intercomparisons/SPICE/SPICE.html).

The Multi-Angle Snowflake Camera (MASC) is a ground-based instrument designed to automatically captures high resolution ($\sim 33.5\,\mathrm{\mu m}$) photographs of falling hydrometeors from three different angles (Garrett et al., 2012). The MASC has been used in previous studies to investigate snowflake properties (Garrett et al., 2015; Grazioli et al., 2017) and to help interpret weather radar measurements (Kennedy et al., 2018). Interestingly, blowing snow particles also trigger the MASC motion detector system (see Section 2.1), producing many images in windy environments. In addition to hydrometeor classification techniques based on MASC images (e.g. Praz et al., 2017), the ability to discriminate between images composed of blowing snow and precipitation particles would therefore be relevant to characterize blowing snow, to provide reference observations to improve its remote sensing, as well as to obtain more accurate snowfall estimates from ground-based sensors. More generally, detailed information about the type of particles extracted from pictures collected by a MASC will enable us to further investigate the complex interactions between wind, snowflakes and snowpack close to the surface in cold and windy regions.

This article presents a new method to automatically determine if an image from the MASC (and potentially other imaging instruments) is composed of blowing snow particles, precipitating hydrometeors (snowflakes and ice crystals) or a mix of both. The classification is accomplished by means of a Gaussian mixture model (GMM) with two components, fitted on a set of

representative MASC images and evaluated on a manually-built validation set. The paper is organized as follows: Section 2 introduces the data sets used to develop the method and fit the GMM. Section 3 illustrates the different steps to isolate the particles and extract related features for the clustering task. Section 4 explains the selection of the most relevant features, the fitting of the GMM as well as the attribution of a flag for mixed images. The main results are shown in Section 5. At last,
limitations and further improvements are discussed in Section 6.

## 2   Instrument and data sets

### 2.1   The Multi-Angle Snowflake Camera

The MASC is a ground-based instrument which automatically takes high-resolution and stereoscopic photographs of hydrometeors in free fall while measuring their fall velocity. Its working mechanism is only summarized hereafter, as more details
and explanations can be found in Garrett et al. (2012), who provide an extensive description of the instrument. Three high-resolution cameras ($2448 \times 2048$ pixels), separated by an angle of $36°$, are attached to a ring structure and form altogether the imaging unit (see Fig. 1). The focal point is located inside the ring at about $10\,\mathrm{cm}$ from each camera (with a focal length of $12.5\,\mathrm{mm}$). Particles falling through the ring and detected by the two horizontally aligned near-infrared emitter-receiver arrays trigger the three flashes and the three cameras. The cameras' apertures and exposure times were adjusted in order to maximize
the contrast on hydrometeor photographs while preventing motion blur effects, leading to a resolution of about $33.5\,\mathrm{\mu m}$ and a sampling area of about $8.3\,\mathrm{cm^2}$ (see Praz et al., 2017). The maximum frequency of triggering is $3\,\mathrm{Hz}$, that is three image triplets per second (see Fig. 6).

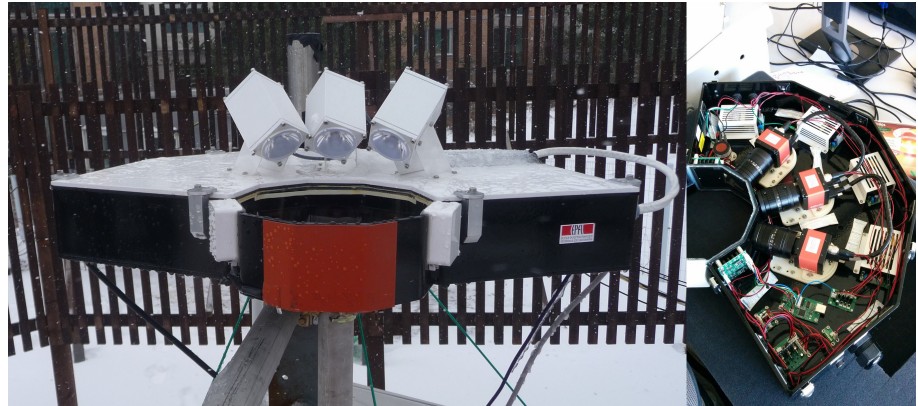

**Figure 1.** Left: side-view of the MASC with the three flash lamps in white on top, the two detectors as white boxes on the side of the metal ring (in black and red in front). Right: top view of the inside of the MASC, with the three cameras clearly visible.

These specifications can be compared to the snow particle counter (SPC) which has been used in many studies of blowing snow (e.g. Nishimura and Nemoto, 2005; Gordon and Taylor, 2009; Kinar and Pomeroy, 2015; Guyomarc'h et al., 2019) and
can be considered as the reference instrument for monitoring blowing snow (e.g. Crivelli et al., 2016). The SPC has a control

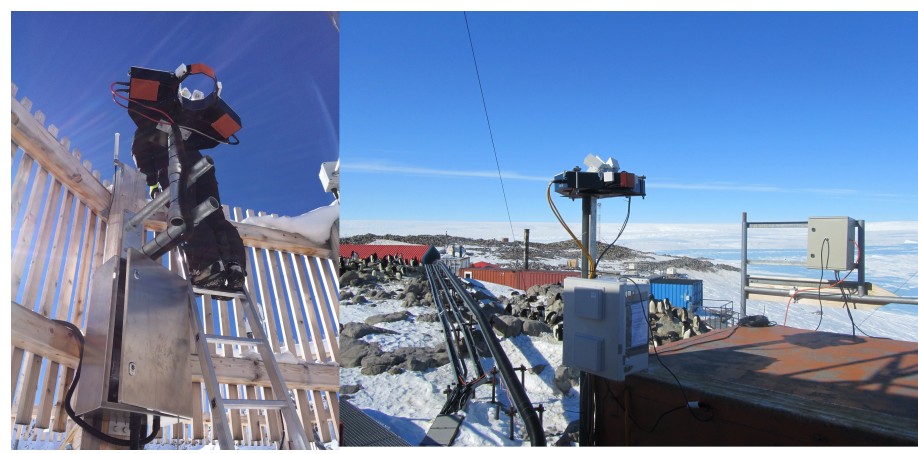

**Figure 2.** Experimental set-up conditions of the MASC in a DFIR near Davos (left) and on top of a container at Dumont d'Urville (right).

volume of $2 \times 25 \times 0.5 \ \mathrm{mm}^3$ and assigns particles into 32 diameter classes between 50 and $500 \ \mu\mathrm{m}$. It provides information on particle diameter (assuming a spherical shape), particle number and particle mass flux usually at a 1-s resolution (but raw data are measured at up to 150 kHz, Nishimura et al., 2014). For more information about the SPC, the reader is referred to the articles mentioned above.

## 2.2 Data sets

The MASC data used to implement and validate the present algorithm were collected during three field campaigns. The first one took place in Davos, Switzerland from October 2015 to June 2016. The MASC was placed at $2540 \ \mathrm{m}$ a.s.l in a Double Fence Intercomparison Reference (DFIR, see Fig. 2, left), designed to limit the adverse effect of wind on the measuring instruments in its center (Goodison et al., 1998). The MASC was about $3 \ \mathrm{m}$ above ground. The two other campaigns took place at the French Antarctic Dumont d'Urville station, on the coast of Adelie Land, from November 2015 to February 2015 and from January to July 2017 in the framework of the Antarctic Precipitation, Remote Sensing from Surface and Space project[1] (Grazioli et al., 2017; Genthon et al., 2018). The instrument was deployed on a rooftop at about $3 \ \mathrm{m}$ above ground (see Fig. 2, right). A collocated weather station and a micro rain radar (MRR) were also installed. Nearly three millions images were collected during these measurement campaigns all together.

From this great amount of data, subsets of pure precipitation and pure blowing snow images were manually selected and further analyzed to choose relevant descriptors and fit a two components GMM. The task of selecting a sufficient number of representative images for both classes turned out to be more complicated than expected, in particular for the Antarctic data set in which mixed images are very frequent. Gossart et al. (2017) used ceilometer data collected at the Neumayer (coastal) and Princess Elizabeth (inland) stations in East Antarctica to investigate blowing snow, and they suggests that more than 90% of blowing snow occurs during synoptic events, usually combined with precipitation. For the sake of generalization,

---

[1]http://apres3.osug.fr

**Table 1.** Campaigns and dates of selected events for the Blowing snow (BS) and Precipitation (P) subsets.

| Antarctica 15-16 | Antarctica 17 | Davos 15-16 |
|---|---|---|
| 11 Nov BS | 08 Feb BS | 23 Feb P |
| 22 Nov P | 09 Feb BS | 25 Feb P |
| 15 Dec P | 18 Feb BS | 04 Mar P |
| 16 Dec P | 19 Feb BS | 05 Mar P |
| 30 Dec P | | 16 Mar P |
| 02 Jan P | | 25 Mar P |
| 11 Jan P | | |
| 28 Jan BS | | |

a large number of representative events was selected across the three campaigns. The goal was to cover a wide range of hydrometeors types and a wide range of snowfall intensities for the precipitation subset. Similarly, a wide range of wind speeds and concentrations were considered to build the blowing snow subset. From the campaigns in Antarctica, pure blowing snow and hydrometeors events were highlighted by comparing time series of MASC image frequency, wind speed and MRR derived rain rate, as illustrated in Figure 3. It was noticed that during strong blowing snow events, the number of images captured by the MASC was much larger than during precipitation events (more than 1 image per second, see Fig. 6). Potential pure blowing snow events were selected when the MASC image frequency and wind speed were higher than their respective median estimated over the whole campaign (to select relatively high values) and no precipitation was detected during the preceding hour. Only events for which these criteria applied for over an hour consecutively were kept. To highlight pure precipitation, the principle was the same but the criteria were an image frequency and a wind speed lower than the median and a MRR precipitation rate greater than zero. The MRR has a certain detection limit, so it was noticed that events selected as blowing snow could also occur during undetected light precipitation. As a result, images from all events were rapidly checked visually and the campaign logbook consulted to ensure that the selection was consistent and coherent. In both cases, some events had to be removed because of obvious mixing of blowing snow and hydrometeors.

As the MASC was deployed inside a DFIR in Davos, no blowing snow events were selected from this campaign. Although the DFIR is supposed to shelter the inner instruments from wind disturbances, we noticed that many images do not solely contain pure hydrometeors. From a webcam monitoring the instrumental set up, we noticed that the fresh snow accumulated on the edges and borders of the wooden structure of the DFIR was frequently blown away towards the sensor. To enlarge the precipitation subset, events with high snowfall rate but not affected by outliers of fresh wind-blown snow were added. Finally, some sparse images of obvious pure hydrometeor in the middle of mixed events were also included in the training set. In total, each subset contained 4263 images and despite possible remaining (limited) uncertainty in the exact type of images, is assumed to be accurate and reliable enough to serve as reference for the evaluation of the proposed technique (see Fig. 8 and Section 4.2).

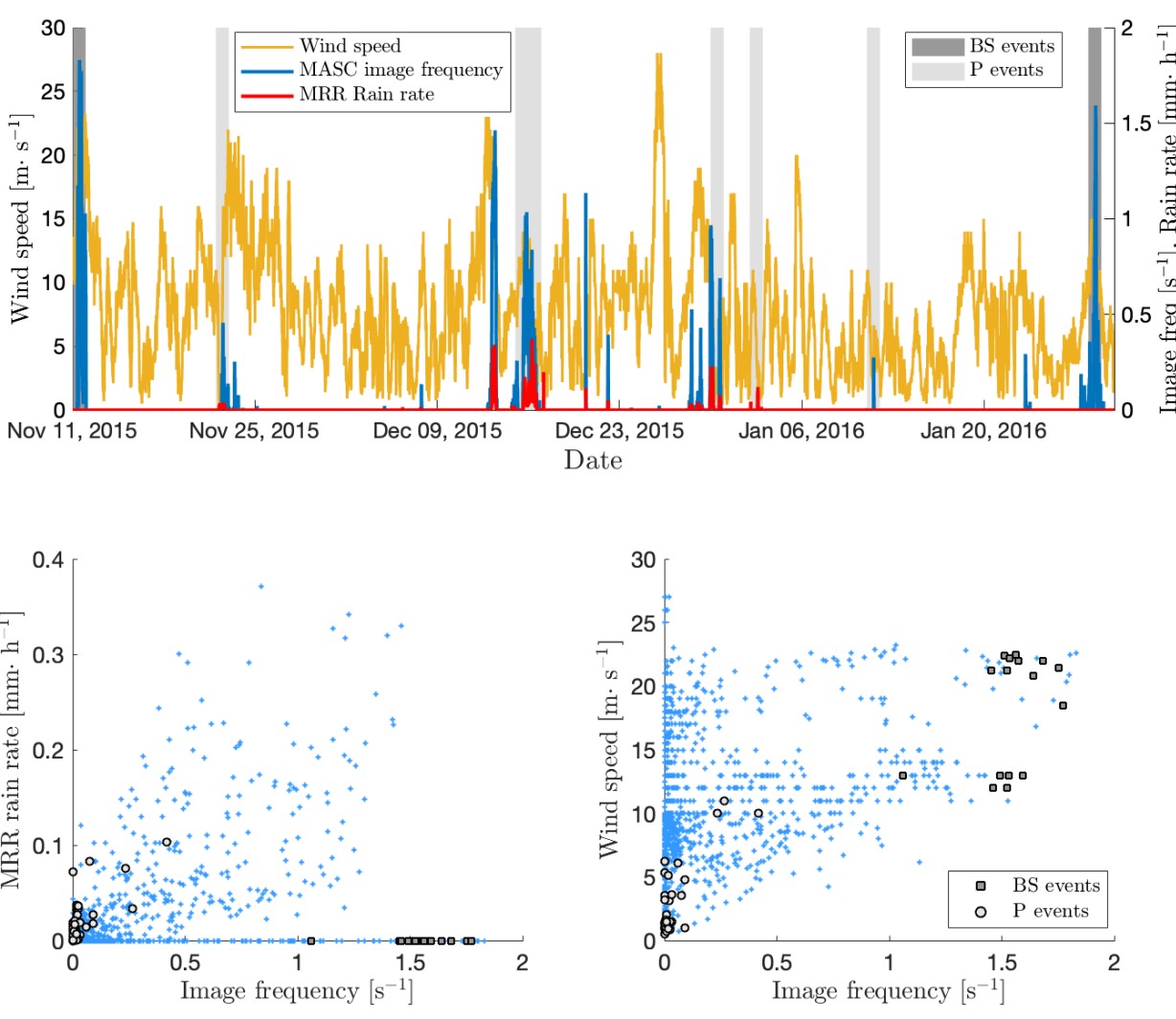

**Figure 3.** Time series and scatter plots of MASC image frequency, wind speed (measured at 10 m) and MRR derived rain rate for the Antarctica 2015-2016 campaign. The grey shading indicates days during which time steps have been selected for the training set as blowing snow (dark grey) or precipitation (light grey). In the bottom scatter plots, the markers figure the selected blowing snow and precipitation time steps. Points on the x-axis in the left scatter plot are potential candidates for pure blowing snow.

## 3    Image Processing

### 3.1    Particle detection

The MASC instrument and the collected images are described in Section 2.1. Although a single particle activates the cameras, many MASC pictures contain multiple particles distributed over the entire image, especially when blowing snow occurs. In fact, the number of particles appearing on a single image is a key characteristic to distinguish between precipitation and blowing snow. As a result, it was deemed essential to detect all particles in each image rather than the triggering one only (which is sometimes unidentifiable). A key challenge of this approach was to get rid of the noisy background. For this purpose, a median filter was used. The brightness of the background strongly depends on the luminosity at the instant of the picture, which varies according to the time of day and can change abruptly in partly cloudy conditions when the sun suddenly appears from behind a cloud. As a result, the median filter shows better performance to remove the background when systematically re-computed over a small number of consecutive images. Assuming that snow particles rarely appear at the exact same position in several consecutive images, the median filter was chosen to be computed over blocks of 5 images per camera angle. To ensure complete removal of the background when its brightness is greater that the corresponding median, a factor of 1.1 was applied to the filter. Finally, as some limited residual noise can still remain in the filtered image, a small detection threshold of 0.02 grayscale intensity was applied to isolate the snow particles. Masks of the sky and reflecting parts of the background (i.e. metallic plates etc) were created for each camera. The multiplication factor and detection threshold are increased in the regions delineated by the masks if the normal filtering leads locally to more pixels detected that one can expect from real particles. These steps are illustrated in Figures 4 and 5. Issues in the filtering may occur if consecutive images are separated by a period of time during which the ambient luminosity has changed significantly (e.g. before/after the sunrise or sunset). An example is shown in Figure B1 in Appendix B.

### 3.2    Feature extraction

Machine learning algorithms require a set of variables, commonly called features or descriptors, upon which the classification is performed. Because of the fragmentation of ice crystals when hitting the snow surface (e.g. Schmidt, 1980; Comola et al., 2017), blowing snow is expected to be characterized by much smaller particle size and much higher particle concentration than snowfall (e.g. Budd, 1966; Budd et al., 1966; Nishimura and Nemoto, 2005; Naaim-Bouvet et al., 2014). In this study, various quantitative descriptors were therefore calculated according to four different categories: the number of particles and their spread across the image, the size of the particles, the geometry of the particles and the frequency at which the images are taken.

Since it is difficult to exactly guess which descriptors are the most adequate to differentiate between blowing snow and precipitation images, an extensive collection of features was extracted from the blowing snow and precipitation subsets and compared. The selection of the most relevant ones is explained in the next section. As the classification is performed at the image level, we need features at the same level and the information on the geometry and size of each detected particle in the considered image must hence be transformed into a single descriptor for that image. Consequently, quantiles ranging from

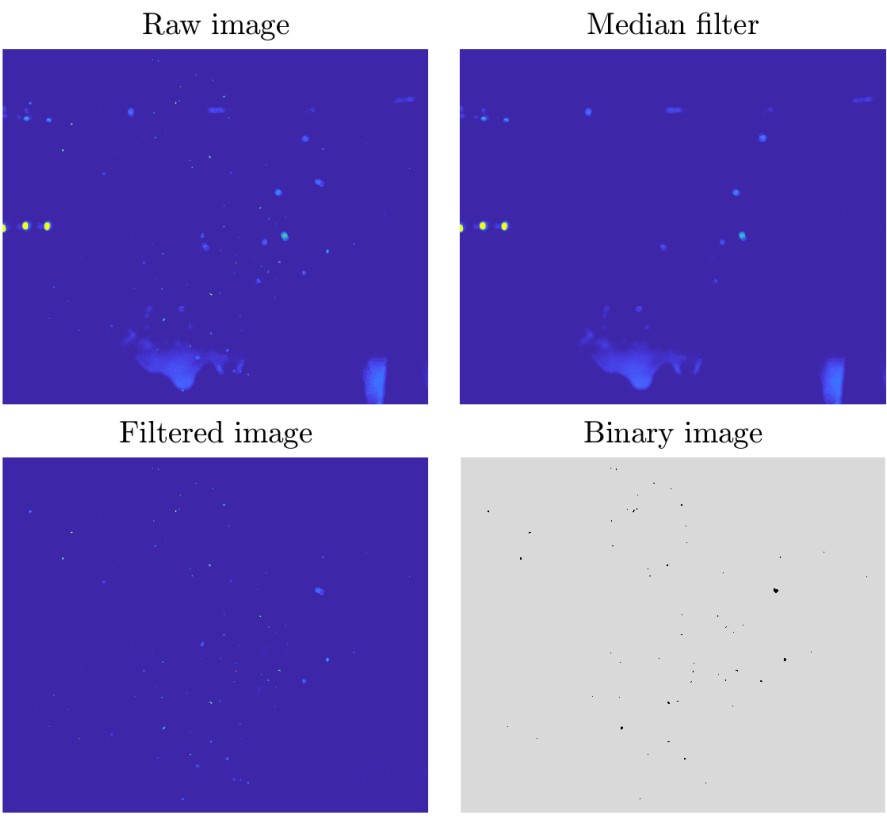

**Figure 4.** Raw image, median filter, filtered image and final binary image for an example of blowing snow particles. The image size is 2448×2048 pixels, corresponding to $82 \times 68.6 \, \text{mm}^2$. Original MASC images are in grey shades, but the color scheme used here aims to enhance contrast and details for visual purposes.

0 to 1 and moments from 1 to 10 were computed out of the distribution of the considered feature within the image. The image frequency is a descriptor independent from the content of the image and thus from the detection of particles. It is therefore not affected by potential image processing issues. As each image comes with its attributed timestamp, the average number of images per minute was calculated with a moving window. The full list of all computed descriptors is displayed in Appendix A. The extraction of features was conducted with the MATLAB Image Processing Toolbox, in particular the function `regionprops`[2].

---

[2]https://ch.mathworks.com/help/images/ref/regionprops.html

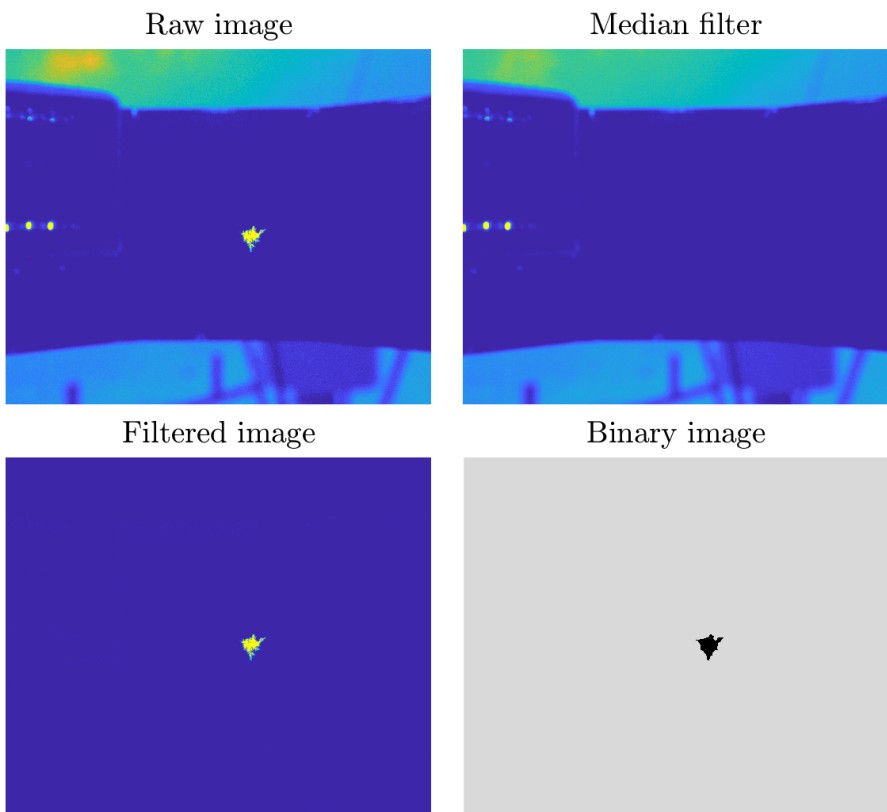

**Figure 5.** Raw image, median filter, filtered image and final binary image for an example of hydrometeor. The color scheme is used to enhance details for visual purposes.

## 4 Classification

### 4.1 Feature selection and transformation

Selecting a relevant set of features and avoiding redundancy is essential for accurate classification, regardless of the classification algorithm. For each of the four categories of descriptors previously mentioned, the most relevant one (according to the criterion explained below) was kept. The descriptor maximizing the "inter-clusters over intra-clusters" distance described in Eq. 1 was selected. This quantity represents the distance between the mean of the blowing snow and precipitation distributions ($\mu_{BS}$ and $\mu_P$ respectively), normalized by the sum of their respective standard deviations ($\sigma_{BS}$ and $\sigma_P$ respectively).

$$S = \frac{|\mu_{BS} - \mu_P|}{\frac{1}{2}(\sigma_{BS} + \sigma_P)}. \tag{1}$$

**Table 2.** Selected features and corresponding S values

| Feature name | S |
| --- | --- |
| Image frequency | 4.43 |
| Cumulative distance transform | 2.89 |
| Maximum diameter quantile 0.7 | 1.71 |
| Squared fractal index quantile 0 | 3.81 |

For the features describing the number of detected particles and their spread across the image, the *cumulative distance transform* was kept. It represents the sum over each entry of the distance transform matrix[3] of the binary image. The distance transform matrix has the same dimensions as the binary image and computes, for each pixel, the Euclidean distance to the nearest 1 element (i.e. the nearest particle). As a result, an image with many particles well distributed over its entire surface will

have a low *cumulative distance transform*, while a single particle, even particularly large, will have a high value. This descriptor is more robust to image processing issues than the raw number of particles, as illustrated in Figure B2 in Appendix B.

Concerning the size distribution of the particles detected in an image, the quantile 0.7 of the maximum diameter was selected (because it has the highest $S$ value among the different quantiles tested). The maximum diameter (*Dmax*) represents the longest segment between two edges of a particle (see Praz et al., 2017, for more details). A logarithmic transformation of this feature

was performed to make the distributions of the two classes more Gaussian. The minimum (i.e. quantile 0) squared fractal index showed the greatest $S$ value (hence discrimination potential) among the features related to the particle geometry indices. The fractal index (FRAC) is defined according to the formula proposed by McGarigal and Marks (1995) in the context of landscape-pattern analysis. It was also more recently used to quantify stand structural complexity from terrestrial laser scans of forests (Ehbrecht et al., 2017).

Due to its different nature, the image frequency descriptor was selected by default, but it is worth noting that it has the highest $S$ value (Eq. 1) among all descriptors (Table 2). The marginal distributions of the selected descriptors for the training set are shown in Figure 6 to provide an idea of their respective magnitude and variability, as well as to illustrate their discrimination potential. As noted above, the image frequency is the most informative descriptor to distinguish blowing snow and precipitation.

In summary, four descriptor categories (related to particle size, particle geometry and particle distribution within the image

as well as image frequency) have been defined to distinguish images collected during blowing snow or snowfall, based on the expected differences in particle size and concentration between the two. A number of descriptors were estimated from each image by computing various quantiles and moments of the distributions of geometric properties of the particles in the considered image. One descriptor from each of the four categories defined above (listed in Table 2) was then selected to be further used for classification as the one maximizing the "inter-clusters over intra-clusters" distance defined in Eq. 1.

[3]https://ch.mathworks.com/help/images/ref/bwdist.html

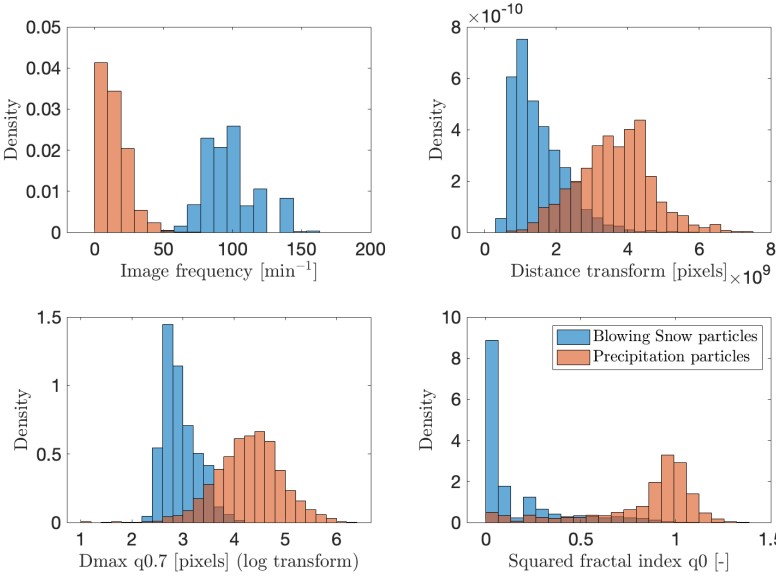

**Figure 6.** Histograms of selected descriptors for the training blowing snow and precipitation subsets.

## 4.2 Model fitting

The choice for the binary classification task was made on a Gaussian mixture model, an unsupervised learning technique that fits a mixture of multivariate Gaussian distributions to the data (see Murphy, 2012; McLachlan and Basford, 1988; Moerland, 2000, for more details). The mathematical description of a multivariate normal distribution is provided in Eq. 2.

$$\mathcal{N}(\boldsymbol{x}|\boldsymbol{\mu},\boldsymbol{\Sigma}) = \frac{1}{(2\pi)^{D/2}|\boldsymbol{\Sigma}|^{1/2}} \exp\{-\frac{1}{2}(\boldsymbol{x}-\boldsymbol{\mu})^{T}\boldsymbol{\Sigma}^{-1}(\boldsymbol{x}-\boldsymbol{\mu})\}. \tag{2}$$

where $\boldsymbol{x}$ is a Gaussian multivariate random variable of dimension $D$, $\boldsymbol{\mu}$ its mean and $\boldsymbol{\Sigma}$ its covariance matrix, with $^{T}$ the transpose operator.

The choice of an unsupervised approach is based on several reasons. First, unsupervised methods do not depend upon labels. Hence, it is not required to ensure correct labeling of each image in the training set. As mentioned earlier, many images are composed of mix of blowing snow and precipitation and it is thus difficult to guarantee the objectivity of all given labels. Second, a clear separation observed between the two subsets would be statistically highly significant as no prior information is provided to the learning algorithm about the classes. Third, for low dimensional problems, unsupervised methods are sometimes less prone to over-fitting and have a better potential of generalization. A main advantage of the GMM compared to other unsupervised methods is to provide posterior probabilities on the cluster assignments and thus allow for soft clustering (i.e. probabilistic assignment). In the context of the present study, this is absolutely relevant as there exists a whole continuum of in-between cases of mixed images. It should be noted that the descriptors were selected using a reference set (see previous section), but the clustering conducted by means of the GMM is itself unsupervised.

A two components GMM with unshared full covariance matrices was thus fitted to the four dimensional ($\boldsymbol{x} = \{f_i\}, i = 1..4$, where $f_i$ are the 4 features listed in Table 2) data composed of the blowing snow and precipitation subsets. The MATLAB Statistics and Machine Learning Toolbox was used for this purpose and the model parameters were estimated by maximum likelihood via the Expectation-Maximization (EM) algorithm[4]. The features were standardized before fitting the model. The mixing weights (or component proportions) were artificially set to $0.5$ by randomly removing 80 data points from the training set and fitting again the GMM to have perfectly balanced classes. This step is essential as the model will then be used to classify new images (possibly from other campaigns). There are no reasons to give more weight to one component, as the relative proportion of blowing snow and precipitation images strongly depends on the campaign location. The posterior probabilities are computed using Bayes rule (Murphy, 2012):

$$P(z_i = k | \boldsymbol{x}_i, \boldsymbol{\theta}) = \frac{P(\boldsymbol{x}_i | z_i = k, \boldsymbol{\theta}) P(z_i = k | \boldsymbol{\theta})}{P(\boldsymbol{x}_i | \boldsymbol{\theta})}, \tag{3}$$

where $z_i$ is a discrete latent variable taking the values $1, ..., K$ and labeling the $K$ Gaussian components. $P(z_i = k | \boldsymbol{x}_i, \boldsymbol{\theta})$ is the posterior probability that point $i$ belongs to cluster $k$ (also known as the "responsibility" of cluster $k$ for point $i$). $P(\boldsymbol{x}_i | z_i = k, \boldsymbol{\theta})$ corresponds to the density of component $k$ at point $i$ (i.e. $\mathcal{N}(\boldsymbol{x}_i | \boldsymbol{\mu}_k, \boldsymbol{\Sigma}_k)$) and $P(z_i = k | \boldsymbol{\theta})$ represents the mixing weight (also denoted $\pi_k$). Note that the $\pi_k$ are positive and sum to 1. $\boldsymbol{\theta}$ refers to the fitted parameters of the mixture model $\{\boldsymbol{\mu}_1, ..., \boldsymbol{\mu}_k, \boldsymbol{\Sigma}_1, ..., \boldsymbol{\Sigma}_K, \pi_1, ..., \pi_K\}$. $P(\boldsymbol{x}_i | \boldsymbol{\theta})$ is the marginal probability at point $i$, which is simply the weighted sum of all component densities:

$$P(\boldsymbol{x}_i | \boldsymbol{\theta}) = \sum_{k=1}^{K} \pi_k \mathcal{N}(\boldsymbol{x}_i | \boldsymbol{\mu}_k, \boldsymbol{\Sigma}_k). \tag{4}$$

As the concern of this study is on two components only, a more compact notation will be used for the rest of the article. The latent variable $z$ will be replaced by $k_P$ and $k_{BS}$ to refer to the precipitation and blowing snow clusters, respectively. The term $\boldsymbol{\theta}$, that denotes the model parameters, will be left implicit. Assuming we are at first interested by performing some hard clustering (i.e. single label to a given image), an image will be classified as blowing snow if $P(k_{BS} | \boldsymbol{x}_i) > P(k_P | \boldsymbol{x}_i)$. That is to say, if the posterior probability to belong to the blowing snow cluster is greater than $0.5$, an image will be classified as such (because the posterior probabilities sum to 1). The model performance was assessed by simply labeling the data points according to its initial subset. An overall accuracy of 99.4% and a Cohen's Kappa score of 98.8% were achieved. The Cohen's Kappa statistic adjusts the accuracy by accounting for correct predictions occurring by chance (Byrt et al., 1993). These high values indicate a very good performance of the fitted GMM. Figure 7 presents the fitted Gaussian components as well as the reference values (not used in the fitting) for each of the 6 possible pairs of the 4 descriptors. It clearly illustrates the performance of the fitted GMM and the discriminative power of the descriptor related to image frequency.

To investigate the stability of the Gaussian components, the precipitation and blowing snow subsets were both randomly permuted and divided in ten equal parts. Ten new training sets of balanced amount of each subset were created and new GMM

---

[4]https://ch.mathworks.com/help/stats/gaussian-mixture-models-2.html

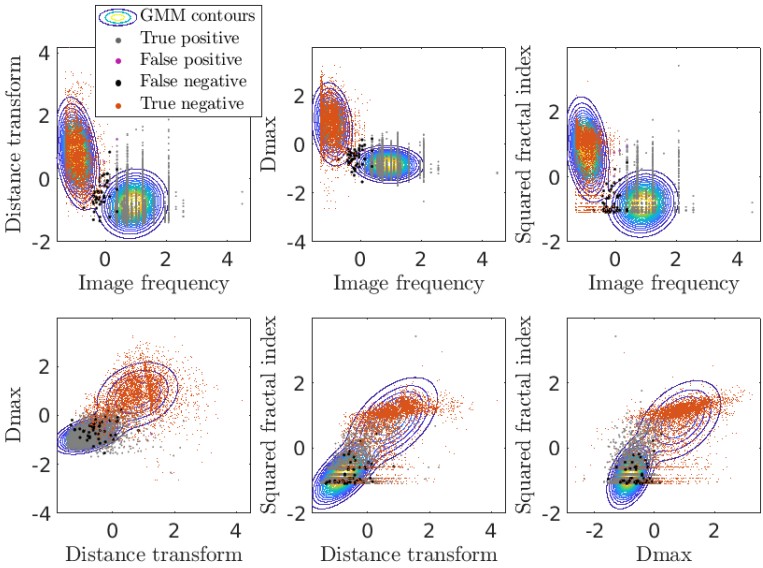

**Figure 7.** GMM contours and data points projected on the 2D planes. The colors correspond to the four entries of the confusion matrix. The predictions result from the clustering and the ground truth is the given labels.

fitted. Figure 8 shows on the top line the boxplots of the Gaussian components parameters $\boldsymbol{\mu}_d$ and $\boldsymbol{\sigma}_d$ (i.e. diagonal entries of $\boldsymbol{\Sigma}$) for each of the four dimensions. The boxplots show a limited variability for each feature (below 10%), indicating a reasonable stability of the fitted parameters. In addition, the bottom line of Figure 8 presents the learning curves, and their fast convergence to the same horizontal line when more than 30% of the training set is used, indicates a data set large enough for a

reliable fitting of the GMM, without overfitting.

### 4.3 Flag for mixed images

As mentioned earlier, an asset of using a GMM model is the posterior probabilistic information that could help estimate the degree of mixing of an image. Data points located close to the decision boundary in the multidimensional space are likely to be composed of a mix of blowing snow particles and hydrometeors. However, distributions of posterior probabilities

computed over thousands of new images from entire campaigns, appeared to be stretched out on both end of the domain (i.e. close to $0$ or $1$) and not many images were present in between. This is probably due to the nature of the descriptors and the resulting shapes and relative positions of the Gaussian distributions. In order to investigate this issue, an additional set of images corresponding to mixed cases was built: it exhibited clear differences in the posterior probabilities with the pure blowing snow and pure precipitation subsets. This differentiation was however around $10^{-6}$ (or $1 - 10^{-6}$), which is not so informative as

such. Consequently, it was decided to define a new index, similar to the posterior probability to belong to the blowing snow component, but more evenly distributed across the range $]0,1[$. The new index uses the negative logarithm of the posterior

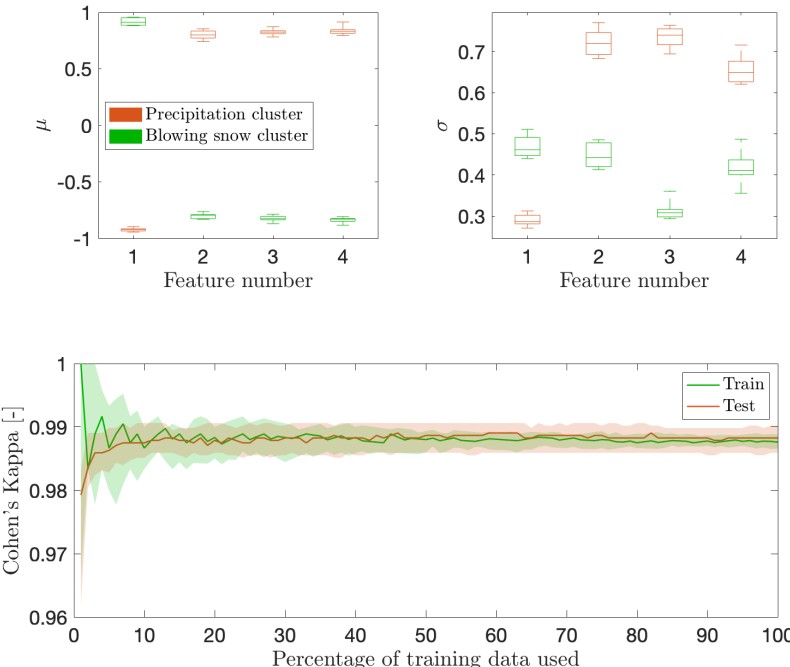

**Figure 8.** Top: stability of the parameters $\mu$ and $\sigma$ (diagonal entries of $\boldsymbol{\Sigma}$) for the two Gaussian components. The boxplots show the distributions of these parameters for each dimension, after fitting the GMM on a 10-fold random split of the training set. The feature number follows the order given in Table 2. Bottom: learning curves for the fitted GMM, showing the evolution of the train and test Cohen's kappa as a function of the proportion of the training samples used. The shaded areas correspond to the 25–75 percentile range computed over 40 iterations of 70-30% random train-test splitting and bold lines are the medians.

probabilities multiplied by the marginal probability. Taking the log of Eq.3 for $k_{BS}$, we have (the same applies for $k_P$):

$$-\log[P(k_{BS}|\boldsymbol{x}_i)P(\boldsymbol{x}_i)] = -\log[P(\boldsymbol{x}_i|k_{BS})P(k_{BS})]. \tag{5}$$

Noting that the term $P(\boldsymbol{x}_i|k_{BS})$ on the right hand side is $\mathcal{N}(\boldsymbol{x}_i|\boldsymbol{\mu}_{BS}, \boldsymbol{\Sigma}_{BS})$, one can substitute Eq. 2 into the above expression, which yields:

5 $$-\log[P(k_{BS}|\boldsymbol{x}_i)] - \log[P(\boldsymbol{x}_i)] = \frac{1}{2}(\boldsymbol{x}_i - \mu_{BS})^T \boldsymbol{\Sigma}_{BS}^{-1}(\boldsymbol{x}_i - \mu_{BS}) + \frac{1}{2}\log(|\boldsymbol{\Sigma}_{BS}|) + \frac{D}{2}\log(2\pi) - \log(P(k_{BS})). \tag{6}$$

The quadratic term on the right hand side is the Mahalanobis Distance, which is a distance that uses a $\Sigma^{-1}$ norm. Hence, it represents the distance between point $\boldsymbol{x}_i$ and the center of the distribution, corrected for correlations and unequal variances in the feature space (De Maesschalck et al., 2000). The second term is related to the determinant of the covariance matrix and equals $-3.94$ for the Blowing Snow component and $-2.59$ for the Precipitation one. The two last terms are constant and sum to

10 $4.37$ (the component proportions were set to $0.5$ and $D = 4$). The right side of Eq. 6 is also known as the quadratic discriminant

function (QDF, Kimura et al., 1987), commonly noted $g_k(\boldsymbol{x}_i)$. The minus in front of the logarithm on the left side of Eq. 6 is used to return positive values and facilitate subsequent graphical interpretations. Note that the constant term $\frac{D}{2}\log(2\pi)$ is often removed, but in this case, it ensures that $g_k(\boldsymbol{x})$ is positive, even for a Mahalanobis distance of zero. Figure 9 displays a scatter plot of the quadratic discriminant values of both components for the whole training set. The proposed index is defined as the

angle of the vector representing a data point on the scatter plot, normalized by $\frac{\pi}{2}$. It is thus computed as follows:

$$\psi = \frac{2}{\pi}\arctan\left\{\frac{-\log[P(k_P|\boldsymbol{x}_i)P(\boldsymbol{x}_i)]}{-\log[P(k_{BS}|\boldsymbol{x}_i)P(\boldsymbol{x}_i)]}\right\}. \tag{7}$$

This normalized angle is bounded in $]0,1[$, with values close to 1 (respectively 0) indicating a strong membership of the considered image (and not the respective proportions within this image) to the Blowing Snow (respectively Precipitation) clusters. It is closely related to the asymmetry of the Mahalanobis distances between a point $\boldsymbol{x}_i$ and the centers of the two

Gaussian distributions, but corrected by the term $\frac{1}{2}\log(|\Sigma|)$ which is different for the two components. The advantage of using the index in this form, rather than deriving it from the Mahalanobis distances alone, is to respect the decision boundary given by the maximum a posteriori (MAP) rule. This means, a posterior probability of $0.5$ yields a $\psi$ index of $0.5$. Finally, quantiles $0.9$ ($\psi_{P0.9}$) and $0.1$ ($\psi_{BS0.1}$) of the $\psi$ index distributions of the points classified as Precipitation and Blowing Snow, respectively, are retained as thresholds to flag potential mixed images. The idea is to allow, for both classes, 10% of the training set images

being flagged as mixed. This value is qualitatively supported by the distribution shown in Figure 9. It can be changed by the user to be more (increasing it) or less (decreasing it) strict on the classification as pure blowing snow or pure precipitation, depending on the intended application.

To provide the user of the method with an easily readable output, a mixing index $\lambda_m$ is introduced by linearly rescaling between 0 and 1 the $\psi$ index of the images flagged as mixed (i.e. $\lambda_m$ is not defined for pure precipitation or pure blowing snow

images):

$$
\begin{aligned}
\lambda_m &= \frac{0.5}{0.5-\psi_{P0.9}}(\psi-\psi_{P0.9}) && \text{if } \psi \in (\psi_{P0.9}, 0.5) \\
&= \frac{0.5}{0.5-\psi_{BS0.1}}((1-\psi_{BS0.1})-\psi) && \text{if } \psi \in [0.5, \psi_{BS0.1})
\end{aligned}
\tag{8}
$$

The mixing index also respects the hard clustering assignment boundary at $0.5$: $\lambda_m > 0.5$ indicates that the image contains a mix of blowing snow and precipitation particles, but overall being closer to blowing snow and vice versa. Images with a normalized angle outside the two mixed thresholds have a NaN index of mixing and are considered as pure blowing snow

particles or pure hydrometeors. Results are provided treating all images independently, but the $\psi$ index can also be averaged among the three camera angles to provide a unique value per image identifier as well. The median of the range (max - min) covered by the $\psi$ values from the three individual views is about $0.08$ in Davos and $0.05$ at Dumont d'Urville, indicating a limited variability between the three views.

In summary, the classification as mixed case is based on the angle characterizing the considered MASC image in the 2D

space formed by the axis related to pure blowing snow on the one hand and the one related to pure precipitation on the other hand. A mixing index $\lambda_m$ is finally computed by linearly rescaling the normalized angle over the range of values corresponding to mixed cases.

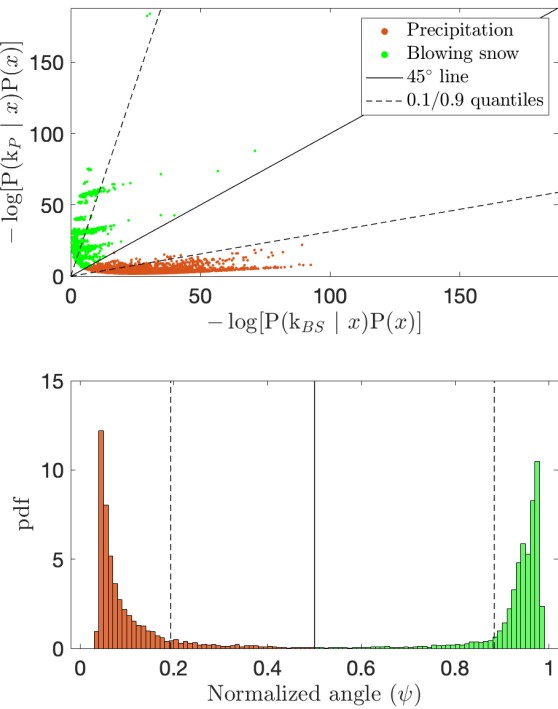

**Figure 9. Top**: scatter plot of the quadratic discriminant values of both components for the training set. **Bottom**: distributions of the normalized angle for the precipitation and blowing snow subsets and thresholds to identify mixed images

## 5 Results

The method presented (and fitted) in the previous sections is now applied to the entire Antarctica 17 campaign (January - July 2017) and to the entire Davos campaign (December 2015 - March 2016). About $2 \cdot 10^6$ images for Antarctica and $8.5 \cdot 10^5$ for Davos were classified. Table 3 summarized the outcome in terms of respective proportions of pure blowing snow, pure precipitation, mixed blowing snow and mixed precipitation, for the Antarctic and Alpine data sets. As expected, the occurrence of blowing snow (pure + mixed) is much more frequent at Dumont d'Urville (75.6%) than at Davos (21.5%, out of which only 0.6% of pure blowing snow).

Figure 10 shows (top) the distribution of the collected MASC images in the space formed by the two quadratic discriminant (one for blowing snow, one for precipitation) as well as (bottom) the distribution of the normalized angle for the entire Antarctica 17 campaign. A clear difference with Figure 9 is the large proportion of values corresponding to mixed cases: there are much more points around the one-one line (top) and a small mode around 0.5 (bottom) for the entire campaign than for the training set (built with much less mixed cases). It is also clear from Figure 10 (bottom) that blowing snow and mixed blowing snow are more frequent than precipitation and mixed precipitation.

**Table 3.** Percentages of MASC images per category

| Class | Antarctica (Jan - Jul 2017) | Davos (Dec 2015 - Mar 2016) |
| --- | --- | --- |
| Pure Blowing snow | 36.5% | 0.6% |
| Pure Precipitation | 7.2% | 39.2% |
| Mixed Blowing snow | 39.1% | 20.9% |
| Mixed Precipitation | 17.2% | 39.3% |

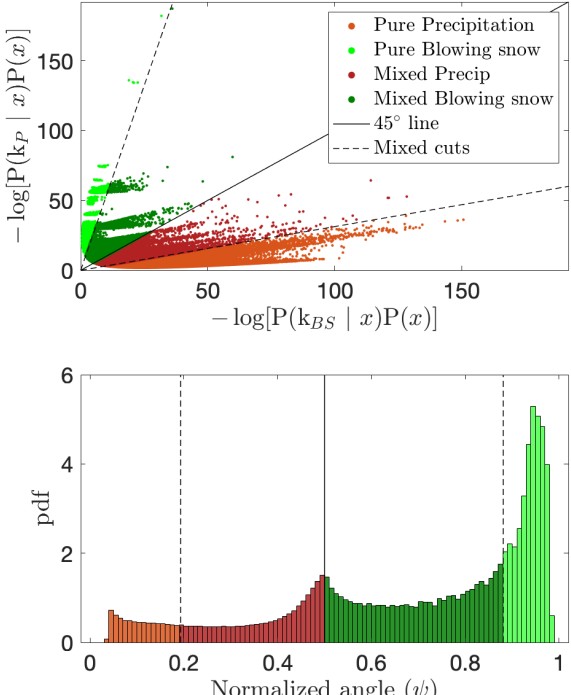

**Figure 10. Top**: scatter plot of the quadratic discriminant values of both components for the entire Antarctica 17 campaign. **Bottom**: distribution of the normalized angle and corresponding classification.

Figure 11 is similar to Figure 10 but for the entire Davos data set. In comparison with Figure 10, the occurrence of precipitation is much larger (and blowing snow much smaller), which is to be expected given the difference in geographic context (Alps vs Antarctica) and experimental set-up (wind-protected vs no wind shield). It should be noted that mixed cases are relatively frequent and that blowing snow still happens in Davos although the MASC was located in a wind shielding fence (DFIR).

5      Beyond global statistics on various data sets as presented above, the proposed approach can also be used to investigate the type of images at high temporal resolutions. Figure 12 shows an example of the output of the algorithm and corresponding

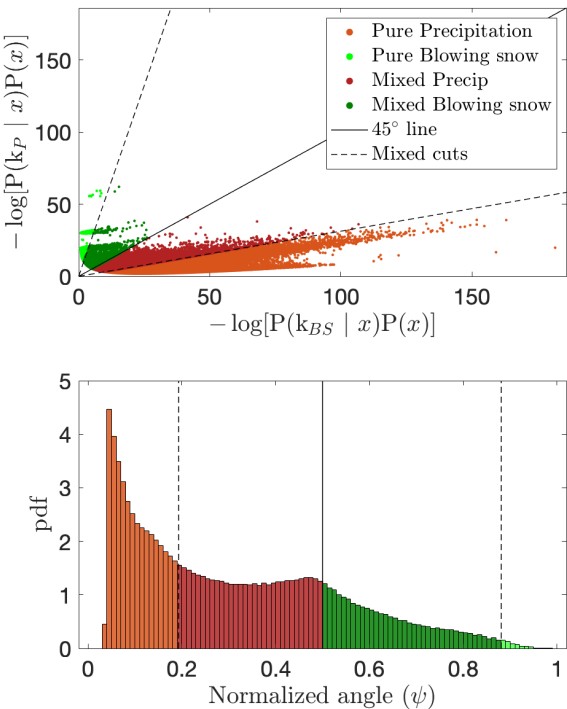

**Figure 11.** Same as Figure 10 for the entire Davos campaign.

images for a few time steps during a mixed event. It illustrates the capability of the proposed approach to distinguish blowing snow, precipitation and mixture in individual MASC images separated by a few seconds (and hence the contribution of the features other than image frequency). Over a longer time period, Figure 13 displays the evolution of the normalized angle for a mixed event during the Antarctica 17 campaign. From roughly 09:00 to 12:00, the types precipitation and mixed are dominant, while between 12:00 and 14:00 the three types (precipitation, mixed, blowing snow) occur simultaneously. This is to be expected at DDU where katabatic winds blow very frequently, even during precipitation (e.g. Vignon et al., 2019). From 14:00 to 22:00, blowing snow becomes dominant (because of stronger winds). After 22:00, mixed cases dominate and some images corresponding to precipitation are detected towards the end of the event. The possibility to identify MASC images corresponding to precipitation, blowing snow or a mixture at a temporal resolution high enough to capture the dynamics of the event is an interesting feature for regions where both are frequently associated.

Considering the full Antarctic and Alpine data sets, it is interesting to analyze the potential differences in their characteristics. Figure 14 presents the distributions of the four descriptors as in Figure 6, but estimated from the entire data sets and not only the training sets (for images classified as pure blowing snow or pure precipitation). It can be seen that while the differences are limited for precipitation (slightly more frequent and larger in Davos than in Dumont d'Urville), they are significant for blowing

| Timestamp | ID | Label | Normalized Angle | Mixing Index |
|---|---|---|---|---|
| 25/03/2016 16:56:35 | 7383 | 1 | 0.7422 | 0.8178 |
| 25/03/2016 16:56:39 | 7384 | 0 | 0.1590 | NaN |
| 25/03/2016 16:56:39 | 7385 | 1 | 0.5659 | 0.5865 |
| 25/03/2016 16:56:40 | 7386 | 0 | 0.1437 | NaN |
| 25/03/2016 16:56:42 | 7387 | 0 | 0.1920 | NaN |
| 25/03/2016 16:56:42 | 7388 | 0 | 0.2260 | 0.0537 |

**Figure 12.** Consecutive MASC images from Davos and their respective classification label, normalized angle and mixing index. Label 1 is for blowing snow. A NaN mixing index means pure hydrometeor (or pure blowing snow). A mixing index close to 1 (top left image) means that it is near pure blowing snow, while a value close to 0 (bottom right image) indicate proximity to pure precipitation.

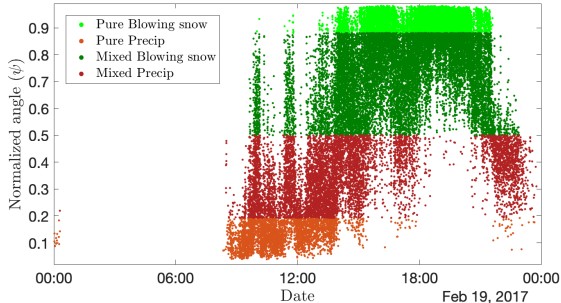

**Figure 13.** Time series of classified MASC images and corresponding $\psi$ values (averaged over the three views) for a mixed event during the Antarctica 17 campaign.

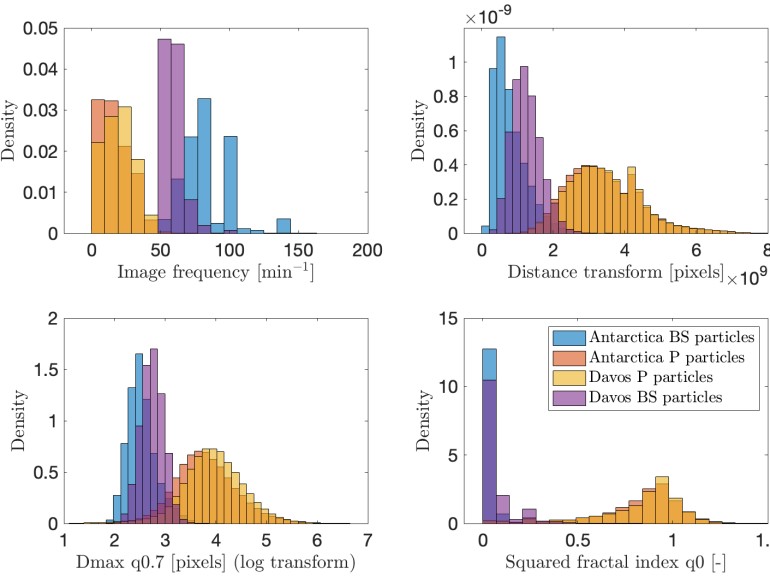

**Figure 14.** Histograms of selected descriptors for the training blowing snow and precipitation images from the entire Dumont d'Urville and Davos data sets.

snow: the blowing snow particles appear less fragmented (larger size and fractal index), less scattered within the images (larger distance transform) and with lower image frequencies in Davos. It should be recalled that the MASC was located in a wind-protecting fence in Davos, so first the occurrence of blowing snow is much smaller (0.6 vs 36.5%), and second it is likely related to fresh snow blown away from the top of the nearby fence.

The MASC resolution ($33.5\,\mu m$) and thresholding (minimum 3 pixels in area) during image processing lead to an image resolution not high enough to capture in full detail the geometry of blowing snow particles. It is nevertheless interesting to plot the distribution of the measured sizes (associated with the MASC sampling area) for blowing snow and precipitation cases and compare it to existing values in the literature. Figure 15 displays the distributions of the measured size (quantified here as $D_{max}$) for blowing snow and precipitation in Antarctica, as well as precipitation in the Swiss Alps. To help visualize the
sometimes overlapping empirical distributions, the fitted Gamma distributions are also plotted. The units are given in [mm], with the approximation that one pixel is $\sim$33.5 [µm].

As expected, the size distribution of blowing snow corresponds to smaller sizes than precipitation: the mode is around 0.2 [mm] for blowing snow and 0.3 to 0.4 [mm] for precipitation. More importantly, the right tail of the distribution is much larger for precipitation than for blowing snow. It should also be noted that the size is slightly larger in the Alpine data set (as
illustrated by the slightly larger mode of the fitted Gamma distributions).

Nishimura and Nemoto (2005) provide size distributions of blowing snow and precipitation measured in Antarctica at Mizuho station using a SPC. The bimodality obtained when combining blowing snow and precipitation data in Figure 15

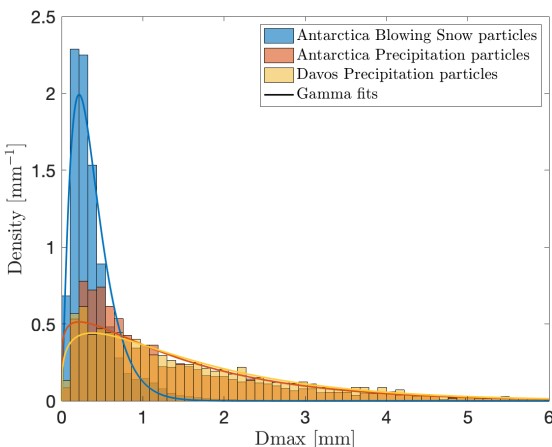

**Figure 15.** Histograms and fitted Gamma distributions of $D_{max}$ for images classified as pure blowing snow and pure hydrometeors from Antarctica and pure precipitation from the Alps.

is in general agreement with the mixed case in their Figure 10. However, the mode for blowing snow appears at a lower size (below 50 [μm] in their Fig.7 at a height of $3.1\,\mathrm{m}$). As mentioned before, this discrepancy is likely due to the limited effective resolution in MASC images after processing. In addition, as there are usually many particles in a single image during blowing snow, some may be out of focus and artificially appear larger than they are. So we expect the blowing snow features extracted from MASC data to be biased towards larger particles. It should also be noted that the sampling areas of the two instruments are different (see Section 2.1) and this could partly explain the differences in the obtained distributions.

Overall, it appears that the MASC images, processed as explained in Praz et al. (2017), are not adapted to a detailed study of the geometry of blowing snow particles, but are still relevant to distinguish blowing snow and precipitation, to characterize mixtures of both and to analyze the dynamics of blowing snow at high temporal resolutions.

## 6 Conclusions

A novel method to automatically detect images from the MASC instrument corresponding to blowing snow is introduced. To classify the images, the method computes four selected descriptors via image processing. The descriptors were selected to be relevant for discriminating between blowing snow particles and hydrometeors as well as to be robust to image processing artifacts. The classification is achieved by a two-component Gaussian mixture model fitted on a subset of $8450$ representative images from field campaigns in Antarctica and Davos, Switzerland. The fitted GMM is shown to reliably distinguish images corresponding to pure blowing snow and pure precipitation cases. The GMM posterior probabilities are also mapped into a new index that allows a better identification of mixed images and a flag signals whether an image is classified as pure hydrometeor, pure blowing snow or mixed. For mixed images, an index between 0 and 1 is proposed to indicate if the image is closer to blowing snow or precipitation. Its evaluation remains qualitative as there are no quantitative observations that can be used as

reference for mixed cases. The outputs are provided for each image independently or for each triplet of images (i.e. information combined over the three cameras of the MASC).

Results from a measurement campaign conducted at the Dumont d'Urville station on the coast of East Antarctica from January to July 2017 suggest that about 75% of the images are affected by blowing snow and that about 36% may be composed of blowing snow particles only (Table 3). The results also suggest that about 56% of the images could be made of a mix of blowing snow and precipitation particles, which support findings that in Antarctica, blowing snow is frequently combined with precipitation (e.g. Gossart et al., 2017). Moreover, time series of the classified images highlight that blowing snow strongly relies upon fresh snow availability and often starts shortly after the beginning of precipitation (Fig.13), which is also consistent with conclusions from Gossart et al. (2017). Results from images taken inside a Double Fence Intercomparison Reference in Davos at $2540\,\mathrm{m}$ a.s.l between December 2015 and March 2016, indicate that despite the sheltering structure, about 60% of the images could be affected to some extent by blowing snow particles from adjacent fence ledges. In terms of percentage of images, these numbers tend to be quite large, as the image frequency is usually much higher when strong blowing snow occurs, but the occurrence is more balanced in terms of time.

As the method was developed and tested on fundamentally different campaigns, it may have a general applicability to any other MASC images. However, it should be noted that some descriptors depends on the particular settings (e.g. image size, pixel resolution) used during the aforementioned campaigns and a new GMM should be fitted if different settings apply. Further work should be conducted to evaluate if the method can give satisfactory results on images that do not include a timestamp, as the image frequency descriptor could not be utilized. In this case, it could be replaced by one or a couple of other descriptors listed in Table A.1 of Appendix A to strengthen the model. The method could also be adjusted to train a model with a supervised learning algorithm that provides posterior probabilities such as Bayesian classifiers or logistic regression. However, this would imply some effort to increase the training set. An inter-comparison between different machine learning algorithms and the creation of different validation sets could help gain confidence in the results.

The main limitations of the present method are the assumption of normally distributed features through the use of the GMM, the too-coarse resolution of the MASC to properly capture the small end of the distribution of blowing snow particle size, and the dependency of the method on the defined training set. The latter illustrates the problem of generalization. Some extremely high intensity snowfall events, higher than the ones observed during the Davos and Antarctica campaigns, could be erroneously classified as blowing snow with the current model due to the nature of the descriptors. In this case, higher intensity pure snowfall events should be included in the training set. Another example is the size of the blowing snow particles. During the campaigns in Antarctica, the MASC was set up on a rooftop at $3\,\mathrm{m}$ a.g.l. Several studies have demonstrated that the size of blowing snow particles tends to decrease with height (Nishimura and Nemoto, 2005; Nishimura et al., 2014). Consequently, blowing snow particles on images from a MASC that would have been set up at much higher or lower heights may have a bias relative to the fitted Gaussian distribution of the Blowing Snow cluster for *Dmax*. It is thus recommended to follow the procedure described in this article and fit a new model, if the one provided does not perform well in other contexts.

## Appendix A: Feature extraction

**Table A1.** Full list of all computed descriptors. Descriptors related to each particle are transformed into a single descriptor for the image (right column). Selected ones are shown with an asterisk

| | |
|---|---|
| Image frequency* | - |
| Number of particles detected in the image | - |
| Distance to connect all particles | - |
| Number of particles smaller than a given threshold | - |
| Ratio of the area represented by all particles to the area of the smallest polygon encircling them | - |
| Cumulative distance transform* | - |
| Maximum diameter* | quantiles 0-1, Moments 1-5 |
| Particle area | quantiles 0-1, Moments 1-5 |
| Particle convex area | quantiles 0-1, Moments 1-5 |
| Particle perimeter | quantiles 0-1, Moments 1-5 |
| Fractal Index (FRAC), Fractal Index squared* | quantiles 0-1, Moments 1-5 |
| Gravelius compactness coefficient (ratio of the perimeter to the one of a circle with equivalent area) | quantiles 0-1, Moments 1-5 |

## Appendix B: Image processing issues

The median filter may perform not satisfactorily, for instance when the background luminosity is changing rapidly (see Fig. B1).

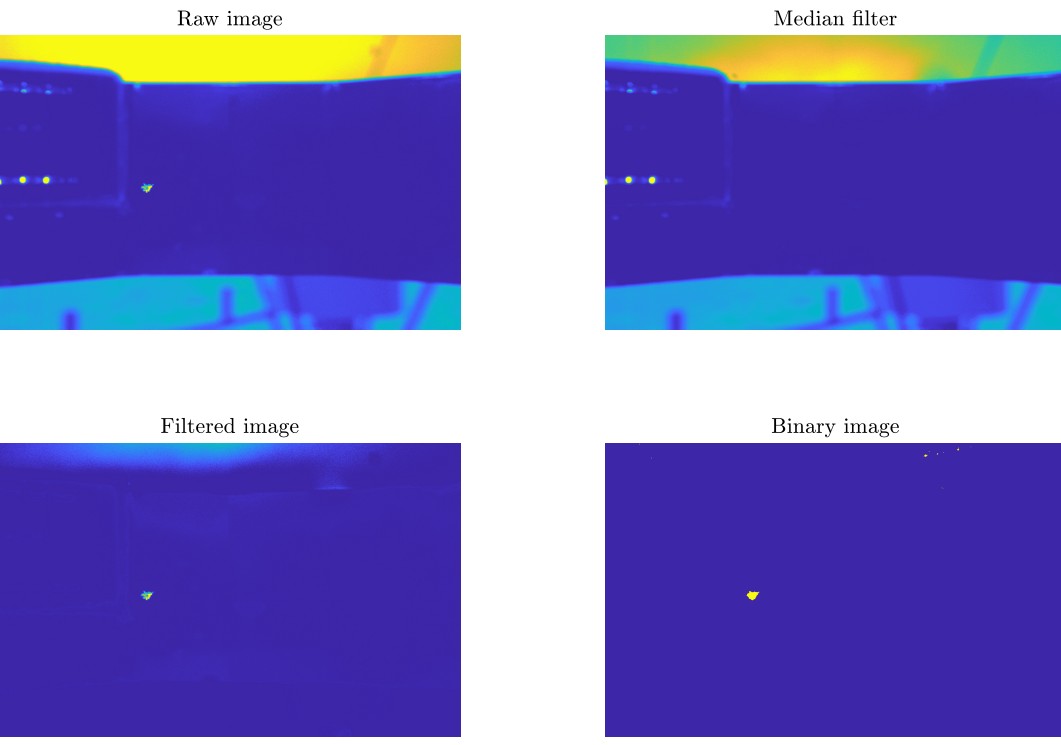

**Figure B1.** Raw image, median filter, filtered image and final binary image for an example where the median filter does not perform well due to changes in sky luminosity. Some artifacts appear on the top right of the binary image

Similarly, large precipitation particles may split or appear as such in the MASC images (see Fig. B2), leading to potential biases in the number of detected particles.

5  *Code availability.* TEXT

*Data availability.* TEXT

*Code and data availability.* The MASC images and the matlab codes used in the present work are available upon request to the authors.

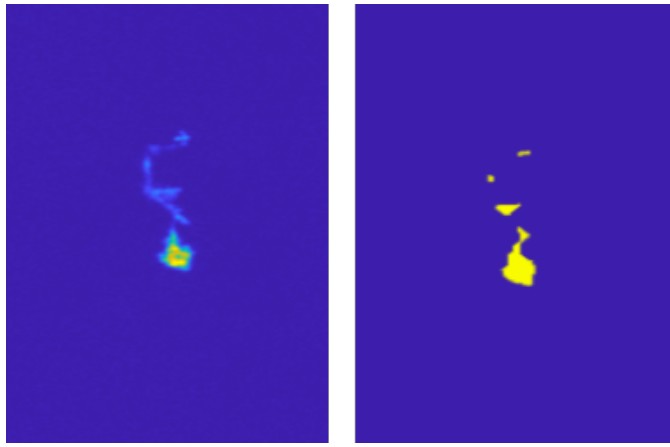

**Figure B2.** A precipitation particle split into fragments that could be confused with blowing snow particles. The Cumulative distance transform descriptor is much less affected by such image processing issues than the number of particles.

*Sample availability.* TEXT

*Author contributions.* TEXT

*Competing interests.* The authors have no competing interests

*Disclaimer.* TEXT

5 *Acknowledgements.* The authors are thankful to Yves-Alain Roulet and Jacques Grandjean from MeteoSwiss, as well as to Claudio Dùran from IGE Grenoble and to the technical staff at the Dumont d'Urville station for their help to collect the set of MASC images used in this study. CP was supported by the Swiss National Science Foundation (grant 200020_175700).

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
