# Peer review of "Identification of blowing snow particles in images from a multi-angle snowflake camera"

_The Cryosphere, 2018_

## Referee Comment (RC1) · Anonymous Referee #1 · 10 Jan 2019

This paper presents a new method to automatically identify blowing snow particles in images from a Multi Angle Snowflake Camera (MASC) that was initially designed to study solid hydrometeors. The author developed an automatic classification task based on four selected descriptors that can make the distinction between blowing snow particles and precipitation particles. Mixed situations are also identified with an index indicating if the image is mostly composed of blowing snow particles or precipitation particles. At the end of their paper the authors present a few examples of application of their method in alpine and polar environment.

The subject of this paper is interesting for the community studying snow and its interactions with the atmosphere in polar and alpine regions. Indeed, in these regions, blowing snow events often occur with concurrent snowfall and the development of innovative

measurements methods is highly relevant to better quantify the interactions between falling snow, the surface of the snowpack and the wind field. My main comments about this study concern (i) the presentation of the feature selection for the classification and (ii) the analysis of results. These questions need to be clarified prior to publication in TC. They are listed below as general comments followed by more specific and technical comments.

General comments

Section 4.1 describes the selection of features used in the classification. The author use four categories of descriptors and mention in Sect. 4.1 which descriptor was finally kept within each category. However, the selection of the descriptors is only qualitatively described and only the final selection is given. The authors should better justify the choice of the descriptors based on quantitative results. Figures 5 and 6 could certainly help but they are never described in the text. It would be also interesting to associate the choice of the final descriptors with physical processes occurring during wind-driven snow transport. For example, the choice of the descriptors related to the size and shape of the particles can be associated with the fragmentation of particles (Comola et al. 2017).

*Comola, F., Kok, J. F., Gaume, J., Paterna, E., Lehning, M. (2017). Fragmentation of wind blown snow crystals. Geophysical Research Letters, 44(9), 4195-4203.*

The authors are presenting the results of their method in Section 5. This section contains 1 table and 5 figures for a total of 9 lines of text. I understand that this paper is centered around the description and evaluation of the identification method but the authors should provide a more exhaustive description and discussion of the results that they decided to show to illustrate the use of their method. For example, Figure 11 is quite interesting and should be analysed more in details by the authors. They could add on this figure the meteorological conditions (wind speed, precipitation) to better explain the transition from a precipitation event to a blowing event. The same apply

to Figure 12. Can the authors comment on the different particle size distribution? For blowing snow particles, how does it compare with particle size distribution measured with Snow Particle Counters (Sato et al., 1993)?

Specific comments

P 2 L 9: present weather have also been used to monitor drifting and blowing snow near the surface (Bellot et al. , 2011).

*Bellot, H., Trouvilliez, A., Naaim-Bouvet, F., Genthon, C., Gallée, H. (2011). Present weather-sensor tests for measuring drifting snow. Annals of Glaciology, 52(58), 176-184*

P 2 L 18-20: Naaim Bouvet et al. (2014) developed a automatic method to estimate the occurrence of snowfall as well as snowfall amount during blowing snow events using measurements from photoelectic sensors. It could be interesting to mention this study in the introduction since it dealt with topics similar to the ones presented in this paper.

*Naaim-Bouvet, F., Bellot, H., Nishimura, K., Genthon, C., Palerme, C., Guyomarc'h, G., Vionnet, V. (2014). Detection of snowfall occurrence during blowing snow events using photoelectric sensors. Cold Regions Science and Technology, 106, 11-21.*

P 4 L 4: the expression "exceptionally important" is rather unclear and the authors should provide typical values of the image frequency during blowing snow events.

P 4 L 15-16: we can expect different properties (size, shape and complexity) for the fresh-wind blown snow particles coming from the edges of the DFIR compared to more classic blown snow particles that have been exposed to transport in saltation and turbulent suspension. Can the author comment about it? In addition, the authors should comment on the potential deposition of blowing snow particles from the surrounding crests. Is it something that can be observed at the experimental site above Davos?

P 6 L3-7: The beginning of Section 3.1 contains a brief description of the MASC. Other technical details are provided at different places in Sections 1, 2 and 3. I recommend

the author to create in Section 2 a sub-section dedicated to the presentation of the MASC and summarizing the main characteristics of the instrument. In this subsection, it would be interesting to add more details regarding the MASC image frequency since it is used by the authors in their image classification method. What is the maximal frequency of the instrument? How does in depend on the particle concentration? To my knowledge, it is the first time the MASC is used to characterize blowing snow particles. It would be interesting if the authors can briefly compare the characteristics of the MASC and the Japanese Snow Particle Counters (SPC) (Sato et al., 1993) in terms of particle characterizations. The SPC can be currently considered as the reference device for blowing snow measurements (fluxes and particle size distribution).

*Sato, T., Kimura, T., Ishimaru, T., Maruyama, T. (1993). Field test of a new snow-particle counter (SPC) system. Annals of Glaciology, 18, 149-154.*

P 6 L 31-33: The authors computed quantiles and moments of the distribution of the considered feature. What are the typical numbers of particles on an image in the different situations: blowing snow, precipitation, and mixed situation?

P 9 L 11: the authors use the term "soft clustering" and the term "hard clustering" at P 10 L 24. These 2 terms can be indirectly understood when reading the text but I recommend the other to include in the text one or two sentences that clearly define these 2 terms.

P 10 L 9: it is not clear why the authors decided to remove exactly 80 data points (images ?) from the training dataset.

P 13 L 9-10: How do the authors justify the choice of having 10

P 13 L 15-17: The authors computed an index of mixing for each image as well as an index average among the 3 images with a given time identifier. Can the authors comment on the variability of the value of the index among the 3 images for a given time? Is their classification methods providing consistent results among the 3 cameras for a

given time? What are the reasons for the potential differences between the images?

P 13 Table 3: as mentioned later in the text (P 15 L 12-13), it would be really relevant for the reader to provide as well the percentages expressed in terms of time. The percentage in terms of images are difficult to interpret since they depends on the image frequency that changes with time.

Technical comments

Text

P 2 L11: the references should be written (Palm et al, 2011) and (Gossart et al., 2017)

P 10 L 4: the signification of the variables used in Eq 2 shoud be given in the text.

Figure

Figure 2: it would be interesting to show the period selected as blowing snow and precipitation on the upper graph of Fig. 2 Maybe add lines showing the median values of wind speed and MASC image frequency that were used to identify the different events.

Figure 2: please indicate at which height above the surface the wind speed is taken.

Figure 3: it is very difficult to identify the blowing snow particles due their size. Could the author insert a zoom over a specific region of the image containing blowing snow particles? It would be also useful to include a scale on the images to allow the reader to better estimate the size of particles.

Figure 4: a scale would be also useful on the images.

Figure 7: the labels and legends on the graphs are hard to read and should be made larger.

Figure 12: mention from which field campaign are taken these data.

Figure B1: It would be interesting to better highlight on the binary image the artifacts

mentioned the caption.

Figure B2: Mention which image is filtered and which one is binary?

---

## Referee Comment (RC2) · Anonymous Referee #2 · 30 Jan 2019

This article introduces a new method to distinguish automatically between blowing snow particles and precipitation particles obtained with MASC. Authors selected four descriptors carefully and classified the images obtained in Davos and Antarctica with utilizing the two components GMM. Then it showed the good performance. Cohen's Kappa score achieved 98.8%; the score of 75% is known to be satisfactory reliable. Further, the mixture of blowing snow and solid precipitation are also discussed. Since the classification of blowing snow and solid precipitation is troublesome, particularly in the strong wind area like in Antarctica. Subjects are topics of conversation on these days and the idea described in this manuscript is also very attractive. Thus, the attitude should be highly evaluated. However, throughout the manuscript, questions arose as shown below. These should be satisfactorily addressed before the paper

can be accepted for the publication. Perhaps it is not crucial, but I am a bit anxious that advanced knowledge of statistics are required to follow the whole contents of this manuscript. In fact, classification procedures are explained considerably careful and I have also learned a bit with the textbook. However, it was still far from satisfactory to follow all of them. They are highly professional and large number of technical terms appears. I appreciate very much if the authors kindly consider the readers who are not so familiar with statistics. Probably it is a good idea to add following two references, such as in the introduction part.. Naaim-Bouvet, F. et al., Detection of snowfall occurrence during blowing snow events using photoelectric sensors, CRST, 106, 11-22, 2014. Nishimura, K., and Nemoto, M., Blowing snow at Mizuho station, Antarctica, Philosophical Transactions of the Royal Soc. of London, A, 363 1647-1662, 2005. The former tried to measure the snowfall amount under the blowing snow condition and the latter showed the snow particle size distribution with SPC and mentions the possibility to detect snowfall. Page 2, Line 6: "The present study focuses on . . . . . . . . . more than 2m above ground." This means when the measurements height is getting lower or much higher, new criteria for classification should be set at each time? Page 2, Line 22: I wonder the resolution of 33.5 ïA∎m is enough to detect the small particles? It is well known that as the position is getting higher, smaller particles will be dominant in the blowing snow. Although the minimum bin in Figure 12 looks 100 ïA∎m, measurement in Antarctica indicates diameter less than 100 ïA∎m shows the maximum even at the height of 10 cm (Nishimura, K., and Nemoto, M.: 2005). Page 2, Line 34: I am a bit anxious whether manually-built validation set is satisfactory accurate. Possible error included should be also discussed. Page 3, Line 13: Same questions listed above. Subsets of pure precipitation and pure blowing snow images were manually selected. I appreciate the authors' efforts very much, but is it perfect? Figure 1: Not only the pictures but also a schematic picture of MASC which shows the basic will be helpful. Page 4, Line 5: Explanations about the MASC should be done before "2. Data sets". Particularly, it should be mentioned here that images are captured when the motion of particles are detected in the field of vision. Otherwise, readers are not able to recognize

the meaning of "MASC image frequency" and the importance as one of the descriptors. Figure 3: I don't understand what do you mean by this figure, probably because I did not follow the procedure precisely. Yellow points shown in "Raw image" and "Median filter" do not corresponds to the blowing snow particles? Then, no particles are shown in the "Binary image"; that is blowing snow particles disappear in the final image. Is this correct? On the other hand, precipitation particles remain in the final image as shown in Figure 4? Figure 4: This figure looks similar to Figure B1. Are there any different meanings? Figure caption of B1 is much clearer than the one of Figure 4. Same explanation should be done in Figure 4 as well. Page 10, Equation (2): Notations of D, T andïĂăïĄ▪ïĂăshould be definedïĂőïĂăI supposeïĂăïĂăïĄ▪ïĂăis not the same as the one in equation (1). Figures 5, 6, 7, 8, 9 and 12; Please make the label of both axes much lager and clearly. It is hard to recognize what is specified respectively. Page 13, line 20: Perhaps/"850'000" should be expressed as "8.5x10ˆ4". Page 13, line 24: Similar particle size distributions are found Nishimura and Nemoto (2005) as well. However, the measurements with SPC revealed that the population of smaller particles than 100 ïĄ▪m shows the maximum. Page 13, line 18; In accordance with the procedures newly introduced in this manuscript, MASC images are classified and results are shown in Figures 9 and 10, and Table 3. Are they reasonable and are there any specific features derived with this analysis? Have you got any new findings? In other words, what sort of contributions you could achieve to the geophysical and cryospherical research field? Or, you would like to remain in just the introduction of the methodology? Figures 6 and 8: No explanations were found in the text. Further, in general, descriptions about the figure are rather brief both in figure captions and text. More detailed explanation is recommended, that will be help to deepen the understandings.

---

## Author Comment (AC1) · 17 May 2019

[12pt]article

[hmargin=2cm,vmargin=2cm]geometry graphicx color natbib

**Identification of blowing snow particles in images from a multi-angle snowflake camera**

*The Cryosphere #tc-2018-248*

M. Schaer, C. Praz and A. Berne

May 17, 2019

**Responses to reviewers**

We would first like to apologize for the delay in preparing the revised version. We then thank the reviewers for their constructive comments which greatly helped improve the manuscript. In the present document, we provide our responses to the comments of the two reviewers. The comments of the reviewers are reported in *italic*, our responses in normal font and the corresponding modifications in the manuscript in blue. Page and line numbers referred to in our responses correspond to the version with changes highlighted.

**Reviewer 1**

General comments

1. *Section 4.1 describes the selection of features used in the classification. The author use four categories of descriptors and mention in Sect. 4.1 which descriptor was finally kept within each category. However, the selection of the descriptors is only qualitatively described and only the final selection is given. The authors should better justify the choice of the descriptors based on quantitative results. Figures 5 and 6 could certainly help but they are never described in the text.*

   Section 4.1 describes how the features are selected using an objective quantitative criterion, but we added text to better explain the approach and the information presented in Fig.5 (p.10, l.15-24):
   The marginal distributions of the selected descriptors for the training set are shown in Figure 6 to provide an idea of their respective magnitude and variability, as well as to illustrate their discrimination potential. As noted above, the image frequency is the most informative descriptor to distinguish blowing snow and precipitation.
   In summary, four descriptor categories (related to particle size, particle geometry and particle distribution within the image as well as image frequency) have been defined to distinguish images collected during blowing snow or snowfall, based on 20 the expected differences in particle size and concentration between the two. A number of descriptors were estimated from each image by computing various quantiles and moments of the distributions of geometric properties of the particles in the considered image. One descriptor from each of the four categories defined above (listed in Table 2) was then selected to be further used for classification as the one maximizing the "inter-clusters over intra-clusters"

distance defined in Eq. 1.

2. *It would be also interesting to associate the choice of the final descriptors with physical processes occurring during wind-driven snow transport. For example, the choice of the descriptors related to the size and shape of the particles can be associated with the fragmentation of particles (Comola et al. 2017). Comola, F., Kok, J. F., Gaume, J., Paterna, E., Lehning, M. (2017). Fragmentation of wind blown snow crystals. Geophysical Research Letters, 44(9), 4195-4203.*

We thanks the reviewer fro this suggestion, that we have added in the text in the beginning of Section 3.2 (p.7, l.27-29):
Because of the fragmentation of ice crystals when hitting the ground surface (e.g. Comola et al., 2017), blowing snow is expected to be characterized by much smaller particle size and much higher particle concentration than snowfall (e.g. Nishimura and Nemoto, 2005; Naaim-Bouvet et al., 2014).

3. *The authors are presenting the results of their method in Section 5. This section contains 1 table and 5 figures for a total of 9 lines of text. I understand that this paper is centered around the description and evaluation of the identification method but the authors should provide a more exhaustive description and discussion of the results that they decided to show to illustrate the use of their method. For example, Figure 11 is quite interesting and should be analyzed more in details by the authors. They could add on this figure the meteorological conditions (wind speed, precipitation) to better explain the transition from a precipitation event to a blowing event.*

We have revised and significantly extended Section 5 to provide more description

and analysis of the results presented in the various figures (see p.16-21).

4. *The same apply to Figure 12. Can the authors comment on the different particle size distribution? For blowing snow particles, how does it compare with particle size distribution measured with Snow Particle Counters (Sato et al., 1993)?*

We have added a few paragraphs at the end of the new Section 5 to discuss the comparison between the size distribution obtained from MASC images and size distributions based on SPC reported in the literature (see p.20-21).

Specific comments

1. *P 2 L 9: present weather have also been used to monitor drifting and blowing snow near the surface (Bellot et al. , 2011).*
   *Bellot, H., Trouvilliez, A., Naaim-Bouvet, F., Genthon, C., Gallée, H. (2011). Present weather-sensor tests for measuring drifting snow. Annals of Glaciology, 52(58), 176-184.*

   We thank the reviewer for this relevant reference, that was added (p.2, l.10).

2. *P 2 L 18-20: Naaim Bouvet et al. (2014) developed a automatic method to estimate the occurrence of snowfall as well as snowfall amount during blowing snow events using measurements from photoelectric sensors. It could be interesting to mention this study in the introduction since it dealt with topics similar to the ones presented in this paper.*
   *Naaim-Bouvet, F., Bellot, H., Nishimura, K., Genthon, C., Palerme, C., Guyomarc'h,G., Vionnet, V. (2014). Detection of snowfall occurrence during blowing*

*snow events using photoelectric sensors. Cold Regions Science and Technology, 106, 11-21.*

We thank the reviewer for this relevant reference, that was added (p.2, l.20 + p.7, l.29).

3. *P 4 L 4: the expression "exceptionally important" is rather unclear and the authors should provide typical values of the image frequency during blowing snow events.*

We modified the sentence to be more clear (p.5, l.5):
It was noticed that during strong blowing snow events, the number of images captured by the MASC was much larger than during precipitation events (more than 1 image per second, see Fig. 6).

4. *P 4 L 15-16: we can expect different properties (size, shape and complexity) for the fresh-wind blown snow particles coming from the edges of the DFIR compared to more classic blown snow particles that have been exposed to transport in saltation and turbulent suspension. Can the author comment about it?*

The new Figure 14 is similar to Figure 7 and presents the distributions of the four selected descriptors, for the entire Antarctic and Alpine data sets, distinguishing pure precipitation and pure blowing snow. These distributions illustrate the differences between blowing snow particles in the two data sets: they appear less fragmented (larger size and fractal index), less scattered within the images (larger distance transform) and with lower image frequencies in Davos than in Dumont d'Urville. This has been added in the text (p.18, l.13 - p.20, l.5):

Considering the full Antarctic and Alpine data sets, it is interesting to analyze the potential differences in their characteristics. Figure 14 presents the distributions of the four descriptors as in Figure 6, but estimated from the entire data sets and not only the training sets. It can be seen that while the differences are limited for precipitation (slightly more frequent and larger in Davos than in Dumont d'Urville), they are significant for blowing snow: the blowing snow particles appear less fragmented (larger size and fractal index), less scattered within the images (larger distance transform) and with lower image frequencies in Davos. It should be recalled that the MASC was located in a wind-protecting fence in Davos, so first the occurrence of blowing snow is much smaller (0.6 vs 36.5%), and second it is likely related to fresh snow blown away from the top of the nearby fence.

5. *In addition, the authors should comment on the potential deposition of blowing snow particles from the surrounding crests. Is it something that can be observed at the experimental site above Davos?*

We discussed with colleagues involved in field measurements at the Davos location, and it is not clear if blowing snow particles from the surroundings could deposit or not into the MASC. So unfortunately, we cannot provide a reliable answer to this question.

6. *P 6 L3-7: The beginning of Section 3.1 contains a brief description of the MASC. Other technical details are provided at different places in Sections 1, 2 and 3. I recommend the author to create in Section 2 a sub-section dedicated to the presentation of the MASC and summarizing the main characteristics of the instrument. In this subsection, it would be interesting to add more details regarding the MASC image frequency since it is used by the authors in their*

[Figure]

*image classification method. What is the maximal frequency of the instrument? How does in depend on the particle concentration? To my knowledge, it is the first time the MASC is used to characterize blowing snow particles.*

A new sub-section (2.1) describing the MASC has been added (p.3-4).

7. *It would be interesting if the authors can briefly compare the characteristics of the MASC and the Japanese Snow Particle Counters (SPC) (Sato et al., 1993) in terms of particle characterizations. The SPC can be currently considered as the reference device for blowing snow measurements (fluxes and particle size distribution).*
*Sato, T., Kimura, T., Ishimaru, T., Maruyama, T. (1993). Field test of a new snow-particle counter (SPC) system. Annals of Glaciology, 18, 149-154.*

See response above.

8. *P 6 L 31-33: The authors computed quantiles and moments of the distribution of the considered feature. What are the typical numbers of particles on an image in the different situations: blowing snow, precipitation, and mixed situation?*

The number of particles on an image is not a descriptor that has been selected for the classification, so it was not analyzed. We can however mention that:

- For precipitation: from 1 particle (ideal case) to several (maybe 10) in case of high intensity snow.
- For blowing snow: from several to one hundred.
- For mixed case: no typical values.

[Figure]

9. *P 9 L 11: the authors use the term "soft clustering" and the term "hard clustering" at P 10 L 24. These 2 terms can be indirectly understood when reading the text but I recommend the other to include in the text one or two sentences that clearly define these 2 terms.*

We have specified the meaning of these two terms (p.11, l.14 and p.12, l.22).

10. *P 10 L 9: it is not clear why the authors decided to remove exactly 80 data points (images?) from the training dataset.*

We removed 80 points to have exactly balanced classes in the training set, as explained on p.12, l.5-9.

11. *P 13 L 9-10: How do the authors justify the choice of having 10%*

The choice of the quantiles 10 and 90% is indeed somehow arbitrary. This means that we assume about 10% of images not corresponding to pure precipitation or blowing snow, which appears reasonable from the bottom plot of Figure 9. It should also be noted that this threshold can be adapted to the user's need/objective. This is now mentioned in the text (p.15, l.17-19):
This value is qualitatively supported by the distribution shown in Figure 9. It can be changed by the user to be more (increasing it) or less (decreasing it) strict on the classification as pure blowing snow or pure precipitation, depending on the intended application.

12. *P 13 L 15-17: The authors computed an index of mixing for each image as well as an index average among the 3 images with a given time identifier. Can the*

*authors comment on the variability of the value of the index among the 3 images for a given time? Is their classification methods providing consistent results among the 3 cameras for a given time? What are the reasons for the potential differences between the images?*

As mentioned on p.15, l.26-27, the values of the normalized angle from the three views can be averaged to characterize a given triplet with a single normalized angle value (used in Figure 13 for instance). In addition, we computed the range (defined as max - min) for each triplet and it appeared limited (median of about 0.08 in Davos and 0.05 in Dumont d'Urville). This was added in the text (p.15, l.27-29):

The median of the range (max - min) covered by the $\psi$ values from the three individual views is about 0.08 in Davos and 0.05 at Dumont d'Urville, indicating a limited variability between the three views.

13. *P 13 Table 3: as mentioned later in the text (P 15 L 12-13), it would be really relevant for the reader to provide as well the percentages expressed in terms of time. The percentage in terms of images are difficult to interpret since they depends on the image frequency that changes with time.*

As the MASC is not regularly sampling in time (falling particles trigger the instrument), there is no direct link between the percentage of images and the percentage of time.

Technical comments

1. *P 2 L11: the references should be written (Palm et al, 2011) and (Gossart et al., 2017)*

Done.

2. *P 10 L 4: the signification of the variables used in Eq 2 should be given in the text.*

   Thanks for spotting the issue, we have added the definition of the variables and also moved equation 2 earlier the text (p.11, l.5-7).

3. *Figure 2: it would be interesting to show the period selected as blowing snow and precipitation on the upper graph of Fig. 2 Maybe add lines showing the median values of wind speed and MASC image frequency that were used to identify the different events.*

   Figure 2 is now Figure 3. In the top plot of Figure 3, we indicated the days (as individual time steps would not have been visible) during which the blowing snow and precipitation images were selected. We also used different markers for blowing snow and precipitation point sin the bottom plots.

4. *Figure 2: please indicate at which height above the surface the wind speed is taken.*

   The anemometer at Dumont d'Urville was at 10 m above the ground. This is now mentioned in the caption of Figure 3.

5. *Figure 3: it is very difficult to identify the blowing snow particles due their size. Could the author insert a zoom over a specific region of the image containing*

*blowing snow particles? It would be also useful to include a scale on the images to allow the reader to better estimate the size of particles.*

Figure 3 is now Figure 4. We have changed the last panel to improve the visibility of the particles in the MASC images. The dimension of the image in pixel and size was been added in the caption.

6. *Figure 4: a scale would be also useful on the images.*

See previous item.

7. *Figure 7: the labels and legends on the graphs are hard to read and should be made larger.*

Figure 7 is now Figure 8. We are sorry for this, the labels and legend font size has been increased.

8. *Figure 12: mention from which field campaign are taken these data.*

Figure 12 is now Figure 15. The data were taken from both locations, as now indicated in the caption.

9. *Figure B1: It would be interesting to better highlight on the binary image the artifacts*

As this figure is in an appendix, we decided to focus on the consistency in the color scheme to better illustrate the different steps of the processing on MASC images.

**Reviewer 2**

1. *Perhaps it is not crucial, but I am a bit anxious that advanced knowledge of statistics are required to follow the whole contents of this manuscript. In fact, classification procedures are explained considerably careful and I have also learned a bit with the textbook. However, it was still far from satisfactory to follow all of them. They are highly professional and large number of technical terms appears. I appreciate very much if the authors kindly consider the readers who are not so familiar with statistics.*

   We have added text in Section 3.2 and Section 4 to better describe and explain the approach for readers who are not experts in classification and machine learning.

2. *Probably it is a good idea to add following two references, such as in the introduction part. Naaim-Bouvet, F. et al., Detection of snowfall occurrence during blowing snow events using photoelectric sensors, CRST, 106, 11-22, 2014. Nishimura, K., and Nemoto, M., Blowing snow at Mizuho station, Antarctica, Philosophical Transactions of the Royal Soc. of London, A, 363 1647-1662, 2005. The former tried to measure the snowfall amount under the blowing snow condition and the latter showed the snow particle size distribution with SPC and mentions the possibility to detect snowfall.*

   We thank the reviewer for these relevant references, that were added (p.2, l.19-20).

3. *Page 2, Line 6: "The present study focuses on ... more than 2m above ground."*
*This means when the measurements height is getting lower or much higher, new criteria for classification should be set at each time?*

As indicated in the conclusion (p.23, l.12-15), there is no guaranty that the fitted GMM (and subsequent classification) is fully relevant for MASC images collected in a potentially very different population of blowing snow particles. We therefore recommend to retrain the algorithm if this is the case.

4. *Page 2, Line 22: I wonder the resolution of 33.5 $\mu$m is enough to detect the small particles? It is well known that as the position is getting higher, smaller particles will be dominant in the blowing snow. Although the minimum bin in Figure 12 looks 100 $\mu$m, measurement in Antarctica indicates diameter less than 100 $\mu$m shows the maximum even at the height of 10 cm (Nishimura, K., and Nemoto, M.: 2005).*

We agree with the reviewer that the MASC resolution (and the image processing) result in the fact that the small blowing snow particles cannot be seen by the MASC. We however think that it is still relevant to be able to distinguish blowing snow and precipitation images at a high temporal resolution. Text has been added at the end of Section 5 to compare size distributions derived from MASC and from SPC (p.19-22).

5. *Page 2, Line 34: I am a bit anxious whether manually-built validation set is satisfactory accurate. Possible error included should be also discussed.*

AS in many applications, the reference data set cannot be totally free of any error. But we did our best to be strict in the manual identification, and this type of

manual classification is a standard approach in machine learning. We added a specific mention of this possible remaining uncertainty in the text (p.5, l.21).

6. *Page 3, Line 13: Same questions listed above. Subsets of pure precipitation and pure blowing snow images were manually selected. I appreciate the authors' efforts very much, but is it perfect?*

See previous.

7. *Figure 1: Not only the pictures but also a schematic picture of MASC which shows the basic will be helpful.*

A subsection presenting the MASC was added (Section 2.1, p.3-4).

8. *Page 4, Line 5: Explanations about the MASC should be done before "2. Data sets". Particularly, it should be mentioned here that images are captured when the motion of particles are detected in the field of vision. Otherwise, readers are not able to recognize the meaning of "MASC image frequency" and the importance as one of the descriptors.*

See previous item.

9. *Figure 3: I don't understand what do you mean by this figure, probably because I did not follow the procedure precisely. Yellow points shown in "Raw image" and "Median filter" do not corresponds to the blowing snow particles? Then, no particles are shown in the "Binary image"; that is blowing snow particles disappear in the final image. Is this correct? On the other hand, precipitation*

[Figure]

*particles remain in the final image as shown in Figure 4?*

We are sorry that the figures were not well readable. We have updated these figures (now Figure 4 and 5) to improve the contrast and make the identified (blowing sow or precipitation) particles more visible.

10. *Figure 4: This figure looks similar to Figure B1. Are there any different meanings? Figure caption of B1 is much clearer than the one of Figure 4. Same explanation should be done in Figure 4 as well.*

Figure B1 illustrates a case for which the median filter is not able to remove all background features because of fast changing background conditions, justifying the use of an additional filtering to obtain the final binary image.

11. *Page 10, Equation (2): Notations of D, T and μ(?) should be defined. I suppose μ(?) is not the same as the one in equation (1). Please note that some of the symbol are not readable in the review.*

Thanks for spotting the issue, we have added the definition of the variables and also moved equation 2 earlier the text (p.11, l.5-7).

12. *Figures 5, 6, 7, 8, 9 and 12; Please make the label of both axes much lager and clearly. It is hard to recognize what is specified respectively.*

We have increased the font size for labels and legends in the different figures.

[Figure]

13. *Page 13, line 20: Perhaps "850'000" should be expressed as "8.5x10$^4$".*

Done.

14. *Page 13, line 24: Similar particle size distributions are found Nishimura and Nemoto (2005) as well. However, the measurements with SPC revealed that the population of smaller particles than 100 $\mu$m shows the maximum.*

We have added text at the end of Section 5 about the comparison with SPC measurements from the literature (p.19-22).

15. *Page 13, line 18; In accordance with the procedures newly introduced in this manuscript, MASC images are classified and results are shown in Figures 9 and 10, and Table 3. Are they reasonable and are there any specific features derived with this analysis? Have you got any new findings? In other words, what sort of contributions you could achieve to the geophysical and cryospherical research field? Or, you would like to remain in just the introduction of the methodology?*

The primary goal of this manuscript is to introduce the proposed methodology to derive information about blowing snow from the MASC. We provide illustration from two contrasted data sets (Alps vs Antarctica) in order to evaluate if the outcome makes sense (and it does!). We also compare the obtained statistics on blowing-snow particle size to reference information from the literature to illustrated the limitations of the proposed approach, mainly related to the too-coarse resolution of the MASC to fully capture the entire size range of blowing snow. This is now clearly mentioned in the text (end of Section 5 and 6).

16. *Figures 6 and 8: No explanations were found in the text. Further, in general, descriptions about the figure are rather brief both in figure captions and text. More detailed explanation is recommended, that will be help to deepen the understandings.*

    We have added text in Sections 4 and 5 to better describe these figures and their analyses.

**Supplement:**

[revised manuscript text omitted]

---

## Author Response (AR2)

**Identification of blowing snow particles in images from a multi-angle snowflake camera**

The Cryosphere #tc-2018-248

M. Schaer, C. Praz and A. Berne

November 21, 2019

**Responses to reviewers (2)**

We thank the reviewers for their constructive comments which helped improve the second revised version of the manuscript. In the present document, we provide our responses to the comments of the two reviewers. The comments of the reviewers are reported in *italic*, our responses in normal font and the corresponding modifications in the manuscript in blue. Page and line numbers referred to in our responses correspond to the version with changes highlighted.

**Reviewer 1**

**General comments**

1. I have read the new version of the manuscript and the answers of the authors to the first round of review. The authors addressed well the main points raised in the first round of review and this improved the quality of the paper. In particular, I enjoyed reading the extended section presenting results that illustrates well the potential of the method developed by the authors. Therefore, I recommend this paper to be accepted for publication in TC. I made below a few comments that the authors should consider prior to publication.

We thank Reviewer 1 for this positive evaluation.

**Technical comments**

1. P4 L 2: SPC are carrying out measurements at frequency higher than 1 HZ. Nishimura et al. (2014) used the high-frequency sampling ability of the SPC to determine the speed of particles during blowing snow events. I recommend the authors to add here something on the high-frequency sampling ability of the SPC.

We have modified the text as follows:

...particle mass flux usually at a 1-s resolution (but raw data are measured at up to 150 kHz,

Nishimura et al., 2014).

2. P7 L 3-7: The characteristics of the MASC has already been described in Section 2.1 and this paragraph could be shortened.

We have shortened the text as follows: The MASC instrument and the collected images are described in Section 2.1.

3. P 7 L 27: replace "ground surface" by "snow surface".

Done.

4. P 13 Fig. 7: The graphs are very hard to read. Indeed, it is hard to identify the False positive and False negative due to their restricted numbers. For some of the plots, the GMM contours are also hardly visible. It would be very good the authors could make this figure easier to read prior to publication.

We have modified Fig.7 to improve its readability. It remains a figure with a lot of information...

5. P 18 L1: The occurrence of blowing snow in alpine terrain presents a strong variability due to the influence of the topography on the atmospheric flow. Results would be certainly different if the MASC had been placing on one of the crests surrounding the Weissflujoch.

We agree with the Reviewer that we expect more blowing snow near the crests in an alpine terrain. Our point here is however to highlight the fact that even within a structure designed to minimize wind effects on solid precipitation (the DFIR), we still see blowing snow occurrence in the MASC data.

6. P 19 Fig 12: Were these images obtained after application of the median filter? The 3 aligned white dots on the left images suggested that it is not the case. If possible, I recommend the authors to show the filtered images here.

We have modified the figure to display the binary images obtained after filtering.

7. P 23, L14: Missing "," between "GMM" and "too coarse".

Added.

**Reviewer 2**

We thank very much Reviewer 2 for their in-depth and detailed evaluation of our manuscript, which helped us clarify important aspects.

**General comments**

1. Primarily, it is as of yet unclear to me whether or not this Gaussian mixture method is necessary or beneficial for categorizing images. Figs 6 and 7, and to some point Figure 14 seem to suggest that the dominate separating factor for the two end-states is the image frequency. Though there is a low finite upper limit on this frequency, this is a loose proxy for snow particle flux in much the same way as a particle counter gives you an index of how many times a sensor is triggered. Physically, it makes sense for the snow flux to differentiate these two regimes as the settling velocity of falling particles is much lower than potential transport speeds, and the potential rates of snow transport by the two methods are quite different. That being said, it would benefit the manuscript greatly if the authors could show that all the technical machinery of the Gaussian mixture model and the addition of the other image analysis metrics (Distance Transform, Squared fractal index, Dmax) are indeed necessary to have accuracy at this order of magnitude and that they are not superfluous technical additions. Other advantages that I may have overlooked would also benefit from being highlighted more.

This is an important point, and we thanks the reviewer for raising it so we can better explain and clarify our approach. As illustrated by the S values in Table 2 and the distributions in Fig. 6, 7 and 14, the image frequency is the most informative feature to distinguish blowing snow and precipitation images. But it must be noted that these values and figures correspond to the training set, composed of selected images of pure blowing snow and pure precipitation. For these "pure" cases, the image frequency would be enough to separate the two types. But when considering all types of images, including pure blowing snow and pure precipitation but also mixture of the two, then the other features contribute to the classification. This is implicitly visible in Fig. 12: the time interval between the top left image (mixed towards blowing snow) and the top right one (pure blowing snow) is about 4 s, 3 s between the top right and the bottom left (pure precipitation), and less than 1 sec between the bottom left and bottom right (mixed towards precipitation). The image frequency is hence similar and even larger for the transition between pure precip and mixed precip. So the difference in the outcome is explained by the other features used for the classification, illustrating their importance for the mixed cases in particular.

We have added the following text in the comment of Fig. 12: (and hence the contribution of the features other than image frequency).

2. I think it would be illuminating for a broader audience and enhance the transparency of the manuscript if it was clearly acknowledged that the normalized angle does not actually give any indication of what proportion of a given image is blowing snow versus precipitation, but actually only indicates what the probability is that an image is one of the two end states according to their training data. As the methods are currently described, this is my understanding of the Gaussian mixture model output. If this is inaccurate, it would also be of benefit to rectify future misunderstandings with further clarification. Furthermore, for technical the paper is,

the validation appears to be largely qualitative, with a tendency towards arguing "typically there is more blowing snow here than there.."

The Reviewer is right in that the normalized angle corresponds to a probability that a given mixed image is closer to pure blowing snow (normalized angle close to 1) or pure precipitation (normalized angle close to 0).

We have added the following text in the description of the normalized angle after Eq.7: of the considered image (and not the respective proportions within this image)

Concerning validation and its qualitative nature, the Reviewer may be confused between the quantitative validation of the fitted GMM performed using the training set (end of Section 4.2, Fig.8) and the application of the proposed method to the entire data sets from the Alps and from Antarctica (Section 5), the evaluation of which is qualitative by essence as we do not have reference data. For the specific aspect of the quantitative evaluation of the mixing index (related to the normalized angle), we similarly do not have reference data to compare to... But it is important to remember that the normalized angle is based on a distance to the cluster centroids, so it is not completely arbitrary.

**Specific comments**

1. P2L4-10: This drifting versus blowing snow designation is unnecessary, and the authors are inconsistent in the use of it. The more technical modes of creep, saltation, and suspensions would be more appropriate differentiations.

We have modified this paragraph into:

Ice particles moving at the snow surface belong to one of the three main types of associated motion: creep, saltatation and suspension (e.g. Kind, 1990). Given the fact that the observations used in the present study were collected about 3 m above the ground (or snow surface) level, the term "blowing snow" hereinafter refers to wind-suspended ice particles.

- P2L18-20: Please refrain from saying obviously as it undervalues the work. We changed to "frequently"
- 3. P2L27: What motion detector system? This has not be referenced yet.

We added "(see Section 2.1)", as the detection system of the MASC is described there.

4. L28-29: How was this adapted, because Praz et al., 2017 says nothing about "blowing snow", "drifting snow", or "fragmented grains".

We changed "combined with" into "In addition to".

- P2L31-33 Unclear motivating statement.
   We have changed into: "...extracted from pictures collected by...".
- 6. P3L17 Does this mean the cameras are not synchronously taking pictures? If so, the sampling frequency is 1 Hz, a distinction of great relevance for blowing snow measurements, where counts scale with flux. This is confusing for Figure 3. What rate is maximal?

The pictures are synchronously taken from the three cameras, at a maximum rate of 3 Hz as can be seen in Figure 6. A reference to Fig.6 has been added at the end of the sentence.

7. P3L20: A better comprehensive reference of (blowing) snow measurement techniques is Kinar and Pomeroy (2015).

Thanks for the reference! We have also added it earlier in the text (3rd paragraph of the introduction).

8. P4L5-13 Refer to the Table, and use the actual months that contained observations (8 days Nov-Jan, etc.), so as to not overstate the amount of data used. Consistently reference the dates (i.e. not just years for one data set and years and months for the other).

Table 1 lists the dates retained to build the training set but the actual duration of the series is much larger. We have clarified the duration of the three data sets.

9. P4L13 Was this 11.5 days total? Please clarify.

We are sorry but we do not understand the question from the reviewer. We hope that the clarification above answers the question.

10. P4L15 Choose not chose.

Changed.

- 11. P4L15 Rephrase "enough". A sufficient number of? Changed.
- 12. *P4L16 classes not class.* Changed.
- 13. P4L16 Especially? How so?

We changed the text into: in particular for the Antarctic data set in which mixed images are very frequent.

14. P4L16 Rephrase "appeared less trivial than expected".

We changed the text into: ...turned out to be more complicated than expected...

15. P4L17 For those that study the cryosphere, but not East Antarctica, how far away are these stations, and are they similar?

Neumayer is on the coast while Princess Elizabeth in about 200 km in land. These stations are mentioned simply because the study by Gossart et al (2017) is using data collected there, but there is no particular importance of their locations for our study. We have added "coastal" after Neumayer and "inland" after Princess Elizabeth in the text to quickly give an idea about the different locations of those two stations.

16. P4L19 "For the sake of generalization..." is not a sentence.

We changed the text into:

For the sake of generalization, a large number of representative events was selected across the three campaigns.

17. P4L20 Correct the phrase "hydrometeors types as well as snowfall rate".

We changed the text into: and a wide range of snowfall intensities

18. P5L5 The sentence beginning with "Similarly" seems like an incomplete or unconnected thought.

We changed the text into: a wide range of wind speeds and concentrations

19. P5L7-9 Why would the image frequency need to be lower than the median during pure precipitation? Is there a physical basis for that?

The median was chosen as a threshold to select values that are high (relatively to their respective distributions), but there is nor physical reason that the values of image frequency associated with precipitation should be below the median. This criterion on the image frequency is used in combination with a similar one for wind and no precipitation during the preceding hour. So all these criteria combined should ensure to select blowing snow. We have added after "their respective median estimated over the whole campaign": (to select relatively high values)

20. P5L14-16 Please be consistent with tenses throughout the paper "we noticed...one could notice".

We changed "one could notice" to "we noticed" to be consistent.

21. *P5L16 Is augment the right word choice here?*

We changed "augment" into "enlarge".

22. P5L19-20 What is this "uncertainty?" Is this 4263 unique instances, or 1421 unique timesteps? Previously commented.

The uncertainty mentioned here corresponds to the uncertainty in the manual labeling of MASC images with particular types. We changed "exact" to "assigned".

There are 4263 images (considering all cameras independently) corresponding to 1421 unique timestamps (triplets). We added (1421 triplets) after "4263 images".

23. P7L6-7 Can you make a mention of the focal length of these cameras? i.e. are all particles always in focus? This is critical for distinguishing blowing snow particles from falling snow.

The focal length is 12.5 mm, but the particles are not always in focus. An empirical quality criterion is proposed in Praz et al., (2017) that can be used to automatically filter out images too much out of focus, but is not used here. The following text has been added (see Section 2.1):

(with a focal length of 12.5 mm)

24. P7L12 What is the window size of your median filter? Median filters have an effective smoothing, depending on window size, essentially blurring all edges.

This median filter is a "temporal" filter and not a spatial filter. The median image is computed over 5 consecutive images, as mentioned on 1.17.

- 25. P7L16 "rarely", not "hardly". "in" not "on". get rid of few or replace with multiple. Changed.
- 26. P7L23 Rephrase "by a too long period of time". We removed "too long".
- 27. P7L27 A more standard and original reference to cite for decomposition of blowing snow grains is Schmidt, 1980 "Threshold wind-speeds and elastic impact in snow transport".We added the reference.
- 28. P7L29 A bit more effort should be made to cite papers where these ideas originated (Budd et al., 1966 "The byrd snow drift project: outline and basic results" and Budd 1966 "The drifting of nonuniform snow particles").

We added the references.

29. P8L1-3 What does that mean? Sentence starting "As..."

We have modified the sentence as follows:

As the classification is performed at the image level, we need features at the same level and the information on the geometry and size of each detected particle in the considered image must hence be transformed into a single descriptor for that image.

30. P9L3 Replace pertinent.

We changed to "relevant".

31. P10L7 This choice seems awfully arbitrary. Can it be backed up by anything?

This choice is based on the S values obtained for different quantiles tested. It is now mentioned in the text.

32. P10L14-15 And what is the significance of having the largest S value?

The distance quantified by the descriptor S (see Eq.1) is selected to rank the different possible features that can be extracted from a 2D images as MASC pictures. We therefore select the features corresponding to the highest S values (indicating that the selected features have the largest discriminative potentials).

33. P10L17 Refers back to my original concern. How do we know all the other machinery surrounding the image frequency is necessary?

See our response to item 1 in the section "General Comments" above.

34. P11L8 You mean many-fold not manifold.

We changed the sentence as follows: The choice of an unsupervised approach is based on several reasons.

35. P12L1 Clear it up and define what the vectors are: "x = (image freq,...)".

We have added:  $(\vec{x} = \{f_i\}, i = 1..4, \text{ where } f_i \text{ are the 4 features listed in Table 2})$  after "four dimensional".

- 36. P12L2 "for this purpose". Changed.
- 37. P12L21 Rephrase "In words" Changed to "That is to say".
- 38. P13L6-7 Where is the degree of mixing actually verified? I only see uncertainty later on, not something physical. Refer to second major comment.

We do not have reference data to verify the degree of mixing (only images corresponding to mixed cases but without an estimation of this degree of mixing). See our response to item 2 in the section "General Comments" above.

39. P13L7-8 This "likelihood" is entirely contingent on your method working. If it does not work, being near the decision boundary means inconclusive.

The mixed cases can only appear at the edges of the GMM peaks, here centered by construction on the two "pure" ends of the spectrum that are clearly separated in the 4-dimension space we fit the GMM in (see Fig.7 for instance). Hence the mixed cases are by construction in between the GMM peaks.

Now the probability we derive is indeed relevant only if the fitted GMM properly describes the empirical joint distribution. We do not have reference data for mixed cases, so we cannot quantitatively evaluate this probability for the mixed cases, but the general trend (conditioned by the GMM) is expected to be correct.

40. P13L11-13 Rephrase sentence beginning with "Nevertheless..."

The sentence has been modified as follows:

In order to investigate this issue, an additional set of images corresponding to mixed cases was built: it exhibited clear differences in the posterior probabilities with the pure blowing snow and pure precipitation subsets.

41. P15L1 Rephrase "The terms have usually opposite signs..."

The sentence has been rephrased as follows: The minus in front of the logarithm on the left side of Eq. 6 is used to return positive values...

42. P16L5-7 Again, how was this generated?

The percentage values provided have been obtained by applying the proposed algorithm, once trained on the specific subsets (see Section 4.2), to the entire data sets at hand (see Section 2.2). We modified the text as follows to clarify this aspect:

The method presented (and fitted) in the previous sections is now applied to...

43. P16L12-13 This is not immediately clear. Precipitation particles are most clearly evident in the top subplot, whereas the blowing snow (combined with mixed?) are dominant below. Please clarify.

The text has been modified as follows:

It is also clear from Figure 10 (bottom) that blowing snow and mixed blowing snow are more frequent than precipitation and mixed precipitation.

44. Fig.10 Why is there a peak around 45 degrees? Does this not imply some tendency towards inconclusive results as the probability is neither one nor the other? I do not recall a training specifically for mixed grains.

This peak near 0.5 for the normalized angle is also visible in Fig.11 (Davos data set) although slightly lower (about 0.47). In addition, there is no such peak in Fig.9 (almost pure blowing snow and pure precip images). This behavior indicates in our view that this peak has a physical basis and is not a pure artifact. As we do not have (quantitative) referenced observations for mixed cases, this unfortunately remains a bit speculative...

45. Fig 11 Are these results? What ground truth do we have for a comparison? These results seem largely qualitative.

Figure 11 displays the outcome of the propose classification method to the entire Davos data set, as Figure 10 does it for the entire Antarctica data set. As the performance of the method has been demonstrated to be good for pure blowing snow and pure precipitation (see Section 4.2), these figures do present results, at least for these two categories. Concerning mixed cases, the results are less quantitative because of the lack of reference data, but still relevant (as for instance the distributions are different between the two regions, in agreement with the respective climatic features) and worth showing in our view.

46. P17L5 Reword sentence beginning with "The proposed..."

We modified the sentence as follows:

Beyond global statistics on various data sets as presented above, the proposed approach can also be used to investigate the type of images at high temporal resolutions.

47. P18L1-2 This seems to imply that you are classifying each cameras images separately. How often did the three cameras agree or disagree in the same few hundredths of a second? This comparison would help bolster the claims that the authors are coming up with self-consistent results. Furthermore, the image classifications self-consistent if one was to remove image frequency? That is, do the other image analysis metrics represent something actually physically relevant, or are the results then "noisy".

About the processing of the three views independently and the consistency of the outcome, this is explained on p.15 (l.25-28). The classification is indeed consistent and marginal differences appear between the three views, showing the robustness of the classification. The mention of hundredths of second was erroneous, and was changed to "a few seconds", as illustrated in Fig.12.

Concerning the added value of the other descriptors than the image frequency, see item 1 in the General Comments above.

48. P18L3-4 Rephrase "the type...and mixed".

The sentence was rephrased as follows: the types precipitation and mixed are dominant

49. P18L5-6 Rephrase the sentence beginning "Finally..."

The sentence was changed to:

After 22:00, mixed cases dominate and some images corresponding to precipitation are detected towards the end of the event

- 50. Fig 12 Really hard to interpret. Please use a binary image or increase the contrast. We have changed Fig.12 and now use the binary images.
- 51. Fig 13 Interesting! And whats the actual truth here? How likely is it that there was "Pure Blowing Snow" happening concurrently with "Pure Precip"?

As with any classification (or model more generally), the truth is hardly known. The concurrent occurrence of blowing snow and precipitation is very likely in Dumont d'Urville, where katabatic winds are very frequent, even during precipitation (Vignon et al., 2019). This is now mentioned in the text.

52. Fig 14 Use more obvious overlapping patterns. It is unclear what is happening when more than two distributions start overlapping.

We did our best but could not find a better way to figure the overlap. In addition, the regions where more than 2 distributions overlap are limited, and although not totally clear, do not hamper the interpretation of the main aspects of this figure.

53. P20L10 Where is Davos blowing snow? If this is blown right off of the fence tops, it should look something like a mix of fresh precip and blowing snow as it has not had a chance to fragment on the ground.

The amount of images corresponding to pure blowing snow in Davos is very limited (0.6%, see Table 3) and deemed not representative, because of the DFIR. So we decided not to plot those as we have a lot of pure blowing snow images from Antarctica.

54. P21L15-16 This has not been convincingly argued.

We have rephrased this paragraph to clarify our results.

55. P21L18-19 This is in effect a methods paper with minimal validation. At the moment, these conclusions are suspect.

We understand the concern about the lack of validation data, for the mixed case only. This being said, the proposed method is thoroughly evaluated for the pure blowing snow and pure precipitation types. The text has been modified to better highlights the strengths and limitations of our approach. But we respectfully disagree with the reviewer when they state that our conclusions are all suspect.

56. P22L1-3 Why not use the actual weather station data nearby, instead of relying on statistics from other years. This reliance makes the conclusions weaker than necessary.

We are not sure to get the point... The reference to Gossart et al. (2017) is used here to show consistency of our results with existing work in the Antarctic environment. And we do not see how we could use standard meteorological observations (without precipitation) to infer the respective occurrence of blowing snow and precipitation.

57. P22L19-20 This should be mentioned much earlier, as there is no reason this assumption should hold.

There is no a priori reason to have the features normally distributed, but the GMM is fitted in order to best match the empirical distribution, by selecting the GMM parameters minimizing the Akaike information criterion (AIC) and the Bayesian information criterion (BIC).

[revised manuscript text omitted]

<sup>1http://apres3.osug.fr